# *Microglena antarctica* sp. nov. a New Antarctic Green Alga from Inexpressible Island (Terra Nova Bay, Ross Sea) Revealed through an Integrative Approach

Riccardo Trentin [1], Enrico Negrisolo [2], Emanuela Moschin [1], Davide Veronese [3], Matteo Cecchetto [4,5] and Isabella Moro [2,*]

1   Department of Biology, University of Padova, Via U. Bassi, 58/B, 35121 Padova, Italy; riccardo.trentin.2@studenti.unipd.it (R.T.); emanuela.moschin@unipd.it (E.M.)
2   Department of Comparative Biomedicine and Food Science, University of Padova, Viale dell' Università 16, 35020 Legnaro, Italy; enrico.negrisolo@unipd.it
3   Department of Pharmaceutical and Pharmacological Sciences, University of Padova, Via Marzolo 5, 35121 Padova, Italy; davide.veronese.4@studenti.unipd.it
4   Department of Earth, Environment and Life Sciences, DISTAV, University of Genova, Corso Europa 26, 16132 Genoa, Italy; matteocecchetto@gmail.com
5   Italian National Antarctic Museum (MNA, Section of Genova), University of Genova, Corso Europa 26, 16132 Genoa, Italy
*   Correspondence: isabella.moro@unipd.it

**Abstract:** One of the aims of the XXXIV Italian Antarctic Expedition is the study of the photosynthetic biodiversity of the Ross Sea. To achieve this goal, sea-ice samples were collected from Inexpressible Island and a strain of a green microalga (IMA076A) was isolated for morphological and molecular investigations. Combining: (1) phylogenetic analyses of the small subunit rDNA (18S rDNA) and of the internal transcribed spacer 2 (ITS-2) sequences; (2) species delimitation methods; (3) comparative analyses of the secondary structures of ITS-2 and compensatory base changes; (4) morphological, ultrastructural and ecological features, we described the strain IMA076A and its relatives as the new species *Microglena antarctica* sp. nov. The discovery of a new species of Chlorophyceae highlights that the biological diversity of Antarctic microalgae is more extensive than previously thought and that molecular phylogeny together with compensatory base changes (CBCs) approach are pivotal in the identification of cryptic microalgae.

**Keywords:** Antarctica; biodiversity; Chlorophyceae; *Microglena*; molecular phylogeny; *Monadina* clade; Ross Sea; 18S rDNA; ITS-2 DNA barcode

## 1. Introduction

Antarctica is considered an immense field laboratory for the study of fundamental global processes with a variety of key organisms that can be used to monitor the functioning of ecosystems, the effects of climate change and the impact of anthropic activities [1]. In particular, Antarctic microalgae play crucial roles in cold ecosystems as inorganic carbon fixers and as keystone species that sustain a high diversity of heterotrophic organisms [2]. Despite the fact that psychrophilic microalgae provide a wide range of ecological functions, many aspects related to their biology, cold adaptations and, especially, their diversity are not well known [2]. In past years, the identification of polar microalgae had been limited to morphological observations [3], which led to the underestimation of the diversity of these photosynthetic organisms [4,5]. However, the use of a polyphasic approach, which combines molecular data, light and electron microscopy, biochemical and physiological analyses, has shed light on the evolution and the diversity of these algae, with the description of novel species and the taxonomic revision of several genera [2,4]. To date, the re-examination of the genus *Chlamydomonas* Ehrenberg [4,6,7] using molecular phylogeny

has highlighted its polyphyletic origin [8–10]. Within this polyphyletic group, the phylogenetic positions of *Chlamydomonas*-like species isolated from snowfields and ice is unclear. According to the taxonomic revision of Demchenko et al., 2012 [4], psychrophilic microalgae isolated in Antarctica form a lineage, the 'Polar' subclade, with the genus *Microglena* Ehrenberg emend. Demchenko, Mikhailyuk & Pröschold [4,6,7], formerly known as the '*Monadina*' clade, which is distinguished from *Chlamydomonas* ('*Reinhardtii*' clade). In this study, we describe a green alga (strain IMA076A) isolated from green sea-ice samples collected from Inexpressible Island (Terra Nova Bay, Ross Sea) during the XXXIV Italian Expedition to Antarctica. We compared strain IMA076A's morphology, ecology and molecular sequences (ITS-2 rDNA and 18S rDNA) with those of three algal strains belonging to the 'Polar' subclade (provisionally described as: *Chlamydomonas pulsatilla* CCCryo 309-06, *Chlamydomonas* sp. ICE-L and *Chlamydomonas* sp. ICE-W) and with data of 21 freshwater and marine strains described as members of the *Microglena* genus [4,11,12]. Particularly, the conserved region of the ITS-2 secondary structure was analysed using the compensatory base changes (CBCs) approach [13]. The analysis of ITS-2 and its corresponding secondary structure is important for the discrimination of biological species of green microalgae since the difference of even one CBC pairing in the conserved region of ITS-2 predicts gametic incompatibility [13,14]. Specifically, the usage of ITS-2 as a barcode marker was adopted to support the delimitation of the genus *Microglena* and to identify its species [4,14]. In an attempt to describe the strain IMA076A, we used an integrated approach based on the phylogenetic analysis using concatenated 18S rDNA and ITS-2 sequences corroborated with morphological and ecological data. In this sense, we propose a new species *Microglena antarctica* Trentin, Negrisolo, Moschin, Veronese, Cecchetto & I. Moro sp. nov., adapted to the extreme Antarctic environment.

Thus, these findings will represent an addition to knowledge on Antarctic biodiversity and will provide a basis for future comparative studies aimed at assessing the multiple effects of environmental changes on the Ross Sea Region.

## 2. Materials and Methods

### 2.1. Isolation and Cultures

Green sea-ice samples were collected with a spatula during the XXXIV Italian Antarctic Expedition (2018/2019), on 18 November 2018, by Isabella Moro and Matteo Cecchetto from the Penguin Lagoon, located at Inexpressible Island (Terra Nova Bay, Ross Sea, Antarctica) with coordinates: 74°54′ S 163°39′ E (Figure 1). In the laboratory, the sea-ice sample, characterized by a green colour was defrosted and the melted ice was streaked with a platinum loop on a plate containing f/2 growth medium [15] solidified with agar. The agar plate was maintained in a growth chamber at 4 °C with a continuous light intensity of ~6 μmol photons m$^{-2}$·s$^{-1}$. In Italy, from the agar plate, the isolation of the microalga, hereforth called strain IMA076A, from the pool of microorganisms present in the plate, occurred using a platinum loop. The material was inoculated in a flask with liquid f/2 growth medium with a salinity of 34‰ to simulate the environmental conditions. Additionally, in this case, the culture flask was maintained in a growth chamber at 4 °C and a continuous light intensity of ~6 μmol photons m$^{-2}$·s$^{-1}$. The liquid cultures of the strain IMA076A were used for the following morphological, ultrastructural and molecular analyses to investigate all the features useful for the identification of the microalga.

### 2.2. Molecular Analysis

#### 2.2.1. DNA Extraction and Amplification of Selected Molecular Markers

Microalgal pellet, obtained by centrifugation of a liquid culture, was ground with mortar, pestle and quartz sand and the DNA was extracted using the DNeasy Powersoil Pro Kit® (Qiagen GmbH, Hilden, Germany), following manufacturer's indications. Prior to the amplification, genomic DNA was quantified with a DU 530 Beckman Coulter UV/v (Beckman Coulter Inc., Fullerton, CA, USA) is spectrophotometer. Two molecular markers (ITS-2 and 18S rDNA) were amplified for further phylogenetic analyses. ITS-2 region was

amplified using the primer pairs ITS1 [16] and ITS4 [17] with the following PCR conditions: 95 °C for 5 min followed by 35 cycles, each including 95 °C for 45 s, 53 °C for 30 s, 72 °C for 45 s, and 72 °C for 5 min. The 18S rDNA region was amplified using the general eukaryotic primers Euk528F, EukA, EukB, Euk1209F and U1391R [18–22] via the following PCR steps: 95 C° for 5 min followed by 35 cycles, each including 95 °C for 45 s, 50 °C for 45 s, 72 °C for 50 s, and 72 °C for 5 min. Amplification products were verified by electrophoresis and purified using HT ExoSAP-IT High-Throughput PCR Product Cleanup reagent (ThermoFisher Scientific, Waltham, MA, USA) before sequencing. PCR products were sequenced at BMR Genomics Sequencing Services with the same primers used for amplification. SeqMan II from Lasergene package (DNASTar, Madison, Wisconsin, USA) was used to create the final consensus sequences, which were compared with those available in online databases by using BLAST [23]. The obtained sequences (1781 bp for the 18S rDNA gene and 1241 bp for the ITS-2 and its flanking regions) were deposited in GenBank, with the following accession numbers: ON185622 and OM791388.

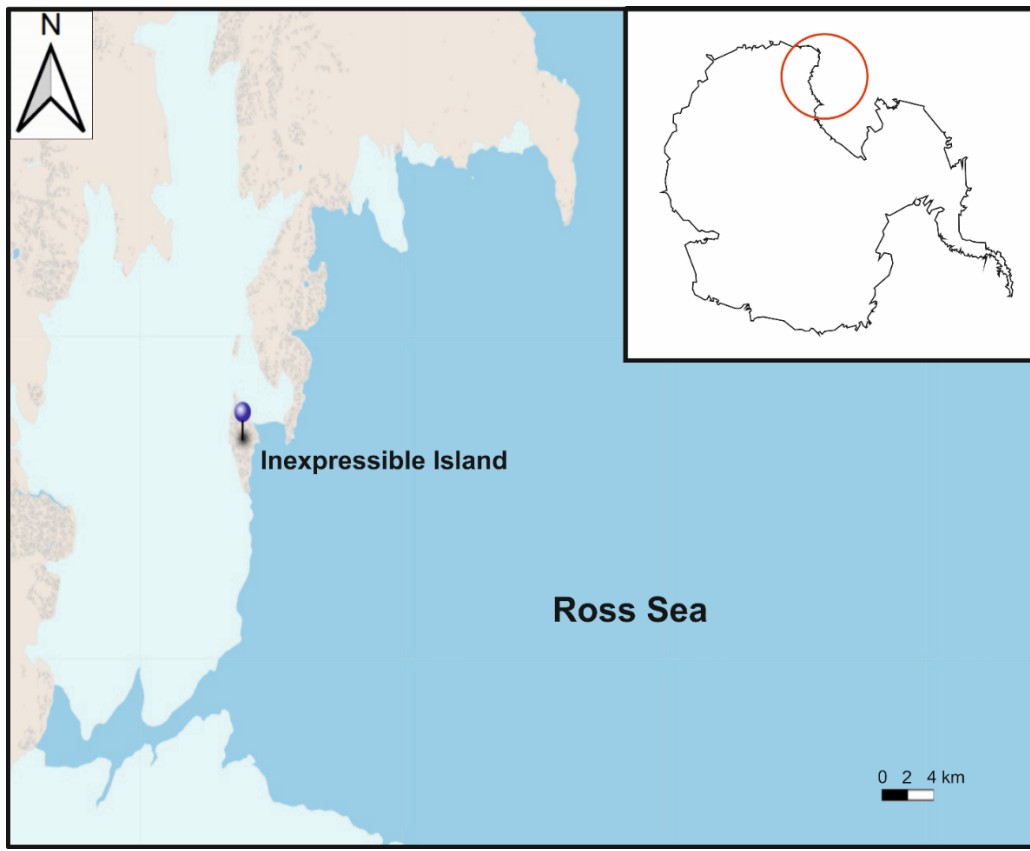

**Figure 1.** Map of Inexpressible Island, the sampling site is defined by the blue pin. The red circle indicates the location of Terra Nova Bay.

2.2.2. Phylogenetic Analyses and Species Delimitation Methods

Separate datasets were created for the 18S rDNA gene and ITS-2 region, including the sequences obtained in this study and other sequences available in the GenBank® database [24] for the genus *Microglena*, for *Wislouchiella planctonica* UTEX 1030, *Phacotus lenticularis* SAG 16.99, *Chaetophora incrassata* CCAP 413/1 and *Aphanochaete magna* UTEX B 1909 (Table S1). These taxa were included in our analysis according to previous phylogenetic studies [4,11,12]. The 29 18S rDNA sequences were aligned with CLUSTALW [25] implemented in MEGA-X 10.2.4 [26]. The obtained alignment (hereafter 18SrDNA.aln) was 1781 positions long. The secondary structures of ITS-2 sequences were initially generated using the UNAfold software [27]. A multiple alignment of 29 ITS-2 sequences (hereafter ITS-2.aln) was produced with the 4SALE program [28] taking into consideration both the

primary sequence and secondary structure. The ITS-2.aln was 779 aligned positions long. The 18SrDNA.aln and ITS-2.aln were successively concatenated in a single dataset (hereafter CONC.aln) in MEGA X 10.2.4. Phylogenetic analyses were performed on CONC.aln according to the maximum likelihood method (ML), implemented in the IQTree 1.6.12 program [29] and to Bayesian Inference (BI) approach available in MrBayes v3.2.7 [30]. In the ML analysis, ModelFinder [31] was used to find the best fitting evolutionary models for each of the two subsets (18S rDNA and ITS-2). The Akaike Information Criterion was used to select the best-fitting evolutionary models. For the 18S rDNA partition, a K2P model [32] with a proportion of invariable sites (I) and a gamma shape parameter (G) was adopted, while for the ITS-2 partition a TPM2u model [33] with empirical base frequencies (F), proportion of invariable sites (I) and a gamma shape parameter (G) was chosen. The robustness of the topologies was assessed by approximate Likelihood Ratio Tests (aLRT) based on Shimodaira-Hasegawa (SH)-like procedures [34], and bootstrap (BT) re-samplings (1000 replicates) in IQTree. The BI analyses consisted of two separate concurrent Markov chain Monte Carlo (MCMC) runs, each composed of four chains (three heated and one cold), for $5 \times 10^6$ generations, sampling trees every 100 generations. At the end of each run, the posterior distribution was considered adequate if the average standard deviation of the split frequencies was $\leq 0.01$. The first 12,500 trees were discarded as burn-in and the consensus topology and posterior probabilities (PP) were derived from the remaining trees. Alternative topologies were evaluated in IQTree using the RELL method [35], Kishino-Hasegawa test [36], Shimodaira-Hasegawa test [37], expected likelihood weights [38] and approximately unbiased (AU) test [39] with 10,000 resamplings. Five species delimitation methods were employed on the ITS-2 dataset: (1) a distance-based method, Automatic Barcode Gap Discovery (ABGD) [40], (2) a tree-based method, Poisson Tree Processes (PTP) [41] and (3) its Bayesian implementation (bPTP), (4) an ultrametric tree-based method, Generalized Mixed Yule Coalescent (GMYC) [42,43] with single threshold (ST) and (5) multi threshold (MT). AGBD was performed directly on the input alignment (ITS-2.aln) using the web interface (https://bioinfo.mnhn.fr/abi/public/abgd/abgdweb.html, accessed on 10 April 2022) with the K80 Kimura distance and relative gap width (X) set to 0.5. PTP and bPTP with 500,000 MCMC generations were run with thinning every 100 generations on the web server (http://species.h-its.org/ptp, accessed on 10 April 2022) using the best maximum likelihood phylogeny tree (constructed in IQTree) as an input. ST and MT GMYC were calculated on the web server (http://species.h-its.org/gmyc, accessed on 10 April 2022) using an ultrametric tree as an input. The ultrametric tree was calculated with BEAST 2.6.3.0 [44] using a GTR+G+T model, a 'log normal relaxed molecular clock' (MCMC: $10 \times 10^6$, sampling every 1000 generations), a 'coalescence tree with constant population', and a 0.25 burn-in. Tree annotator 1.8.0 [45] from the BEAST package was used to build the consensus of sampled trees.

### 2.2.3. DNA Barcoding, CBCs/p-Distances

ITS-2 sequences of the species belonging to the former '*Monadina* clade' were aligned in MARNA [46] and the conserved regions were extracted manually from this alignment. According to Coleman instructions [47], conservative ITS-2 regions were used to find species-specific molecular differences among species. The regions selected for DNA barcode consisted of the first 15 bp in 5.8S-LSU stem, the first 5 bp of Helix 1, the first 11 bp of Helix II and the alignable base-pairs of Helix III. The extracted portions were aligned again in MARNA. Base-pairs alignments among all sequences were converted by numbers according to different base pairings (1 = A-U 3 = G-C 5 = G•U 7 = mismatch 2 = U-A 4 = C-G 6 = U•G 8 = deletion, unpaired of single bases) using R-Statistics® 3.5.3 version. Sequence alignment was used to detect compensatory base changes (CBCs) using CBCAnalyzer version 1.1 [48]. Uncorrected p-distances among species were calculated using PAUP [49] and the resulting matrix was visualized using a non-metric multidimensional scaling (NMDS) calculated with the 'vegan' package implemented in R [50]. Results of these analyses were used to create a CBC/p-distances matrix.

### 2.3. Morphological Analysis

The isolate was studied under differential interference contrast (DIC) microscopy with a Leitz DM IRB microscope (Leica, Wetzlar, Germany) and a Zeiss LSM 700 confocal microscope (ZEISS, Jena, Germany) equipped with a digital image acquisition system. To prepare samples for SEM microscopy, cultured cells were fixed in glutaraldehyde 2.5% in 0.1 M cacodylate buffer (pH 6.9) and then post fixed in 1% OsO4 in the same buffer for 2 h. Samples were dehydrated in a graded concentration increasing ethanol series with centrifugation following every different concentration step. After these steps, ethanol was removed by Critical Point Drying and finally samples were gold coated. SEM observations were carried out using a scanning electron microscope FEI Quanta 200 variable pressure-environmental/ESEM (FEI, Eindhoven, Netherlands) working at a working distance of 13.9 mm and at a 20 kV voltage. For TEM analyses, samples after post fixation were dehydrated in ethanol as above and treated in propylene oxide to be then included in araldite and maintained at 60° for 72 h in order to polymerize the resin. Ultrastructural analyses were carried out through a FEI Tecnai G2 (FEI, Eindhoven, Netherlands) transmission electron microscope, equipped with a side mounted camera Olympus Veleta (Olympus, Münster, Germany) and a bottom mounted camera TVIPS F114 (TVIPS, Gauting, Germany). Morpho-ecological data for strain IMA076A and phylogenetically related species were summarized in a matrix and visualized using a principal component map calculated with the 'PCAmixdata' package implemented in R.

## 3. Results

### 3.1. Phylogeny and Species Delimitation Methods

High-quality sequences were obtained for the two markers amplified from the newly studied green microalga strain IMA076A. We amplified 1781 bp for the 18S rDNA gene and 1241 bp for the ITS-2 and its flanking regions. Phylogenetic analyses performed on the CONC.aln, based on ML and BI approaches, provided identical topologies (Figure 2). The best unconstrained tree showed three distinct and well-supported lineages, two marine clades: 'clade I' (including polar-strains IMA076A, CCCryo 309-06, ICE-L and ICE-W and *M. redcarensis* SAG 18.89), 'clade II' (including *Microglena reginae* SAG 17.89 and *M. uva-maris* SAG 19.89) and one freshwater lineage 'clade III'. In the best tree, strain IMA076A grouped with three strains of Antarctic green algae: CCCryo 309-06, ICE-L and ICE-W. This grouping received strong statistical support: 99% SH–aLRT value, 1 posterior probability, 100% bootstrap. This lineage formed a well-supported clade (100% SH-aLRT value, 1 posterior probability, 100% bootstrap) with the marine species described by Nakada et al., 2018 [11] as *M. redcarensis* SAG 18.89. This last clade was sister taxon to the other species of the genus *Microglena*. Five species delimitation methods were used to test if the polar-strains IMA076A, CCCryo 309-06, ICE-L and ICE-W belong to the same species (Figure 2). All the species delimitation methods (PTP, bPTP, ABGD, ST-GMYC and MT-GMYC) showed unequivocally that the polar strains belong to a single species, which is closely related to the strain SAG 18.89.

### 3.2. Hypothesis Testing

The tree constrained to monophyly *M. redcarensis* SAG 18.89 with *Microglena reginae* SAG 17.89 and *M. uva-maris* SAG 19.89 (clade II) was significantly worse than the best unconstrained tree (Figure 2), as was the tree constrained the monophyly of *M. antarctica* (IMA076A, CCCryo 309-06, ICE-L and ICE-W) and *M. redcarensis* SAG 18.89 (clade I) with freshwater species (clade III). A tree constrained to the marine strains (*M. antarctica* IMA076A, CCCryo 309-06, ICE-L and ICE-W, *M. redcarensis* SAG 18.89, *Microglena reginae* SAG 17.89 and *M. uva-maris* SAG 19.89) to monophyly was not significantly different from the best unconstrained tree found (Table 1).

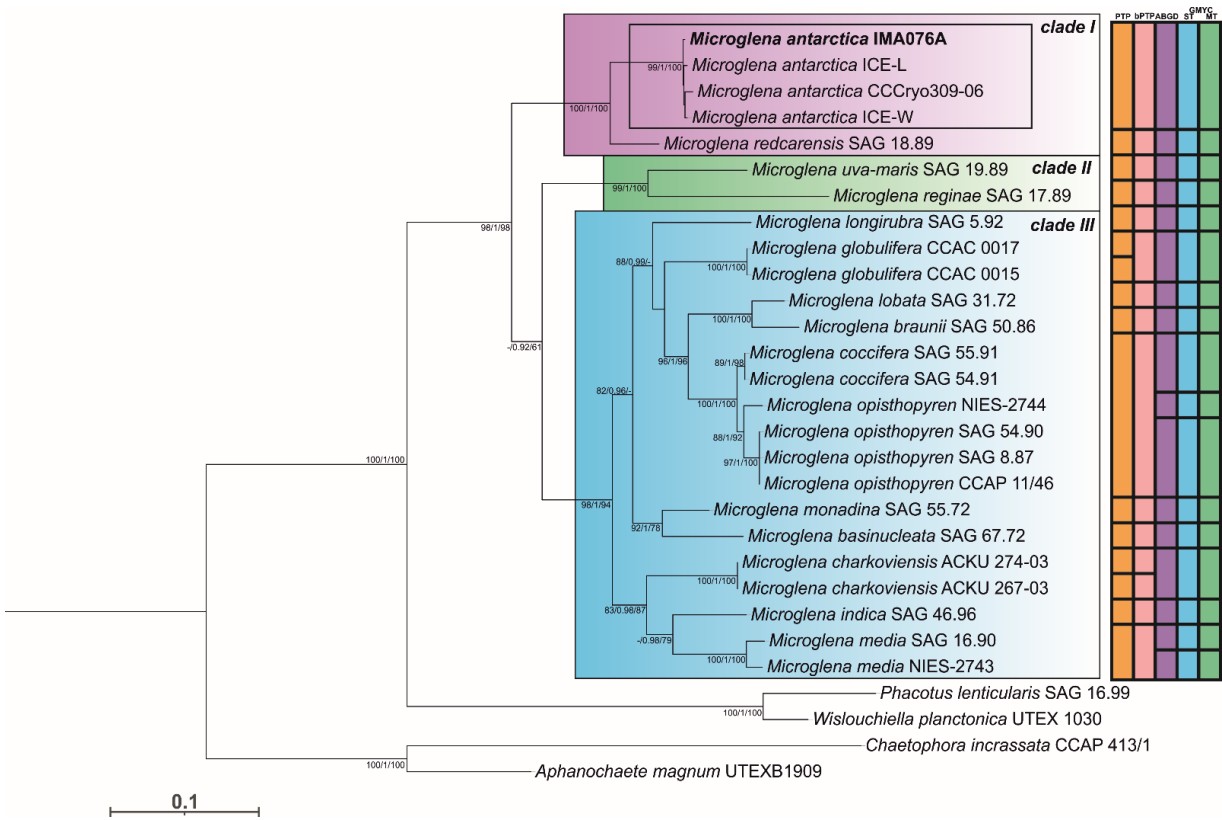

**Figure 2.** Maximum likelihood (ML) phylogeny of concatenated the 18S rDNA and ITS-2 regions alignment with members of the genus *Microglena*. Approximate Likelihood Ratio Tests based on Shimodaira-Hasegawa-like procedures (SH-aLRT) values (%), Bayesian Posterior Probabilities (PP) and ML bootstrap values (%) are shown above branches (SH-aLRT supports ≥80%, posterior probabilities ≥0.70 and bootstrap values ≥50%). Results of species delimitation methods: PTP = Poisson-Tree Processes and bPTP (Bayesian version), ABGD = Automatic Barcode Gap Discovery, GMYC = Generalized Mixed Yule Coalescent (ST: single threshold and MT: multiple threshold) are summarized on the right side of the tree.

**Table 1.** Results of topology test. Scores of constrained trees: (BT) = Best unconstrained tree, (Clade I + II) = *M. uva-maris*, *M. reginae*, *M. recarensis* and *M. antarctica*. (Clade I + III) = *M. recarensis*, *M. antarctica* and freshwater *Microglena* species and (Clade II + R) = *M. redcarensis*, *M. uva-maris* and *M. reginae.* In each case, all exemplars within the constraint were initially assumed to be unresolved with respect to one another. deltaL: logL difference from the maximal logl in the set; bp-RELL: bootstrap proportion using RELL method; p-KH: *p*-value of one-sided Kishino-Hasegawa test; p-SH: *p*-value of Shimodaira-Hasegawa test; p-WKH: *p*-value of weighted KH test; p-WSH: *p*-value of weighted SH test; c-ELW: Expected Likelihood Weight; p-AU: *p*-value of approximately unbiased (AU) test. Plus signs denote the 95% confidence sets. Minus signs denote significant exclusion. All tests performed 10,000 resamplings using the RELL method.

| Tree | logl | delta | bp-RELL | p-KH | p-SH | p-WKH | p-WSH | c-ELW | p-AU |
|---|---|---|---|---|---|---|---|---|---|
| BT | −9255.53 | 0 | 0.625+ | 0.645+ | 1+ | 0.645+ | 0.881+ | 0.62+ | 0.648+ |
| Clade I+II | −9257.77 | 2.2402 | 0.349+ | 0.354+ | 0.803+ | 0.354+ | 0.782+ | 0.348+ | 0.483+ |
| Clade I+III | −9320.01 | 64.485 | 0− | 0.0001− | 0.0002− | 0.0001− | 0.0001− | 0.0002 | 0.0398− |
| Clade II+R | −9266.84 | 11.313 | 0.0263− | 0.12+ | 0.321+ | 0.0578+ | 0.108+ | 0.0326− | 0.0355− |

### 3.3. ITS-2 Secondary Structure, DNA Barcoding, CBCs/p-Distances

The secondary structure analyses of the ITS-2 rDNA (Figure 3) revealed a high similarity between strain IMA076A and polar strains (CCCryo 309-06, ICE-L and ICE-W) and

remarkable differences in III and IV helices between IMA076A and *Microglena* species. Helix III in IMA076A consisted of 228 bases, as in polar-strains CCCryo 309-06, ICE-L and ICE-W, and resulted longer compared to helix III of the species of the genus *Microglena*. Interestingly, while the proximal region of the helix was conserved among *Microglena* species, the distal region represented an expansion detectable only in polar strains. Helix IV was 80 bases long in IMA076A and ICE-L, while it consisted of 78 bases in ICE-W and CCCryo 309-06. In contrast, *Microglena* species have reduced IV helices, usually less than 30 bases. Our results highlighted that p-distances among polar strains were equal to zero, and ITS-2 sequences of polar strains did not show CBCs in the barcode-alignable region (Table 2), which supports the view, with a reasonable probability (about 76%), that these microalgae belong to the same species [13]. The marine strain SAG 18.89, which resulted the sister species to the polar algae (Figure 2), noticeably showed the lowest values in genetic distance and CBCs differences with IMA076A with respect to all the other *Microglena* species (Table S2). In particular, the NMDS plot of p-distances (Figure 4) showed that the closest strains to the Antarctic clade species (ICE-L, ICE-W, CCCryo 309-06 and IMA076A) were SAG 18.89, SAG 17.89 and SAG 19.89, which were clearly separate from other *Microglena* species.

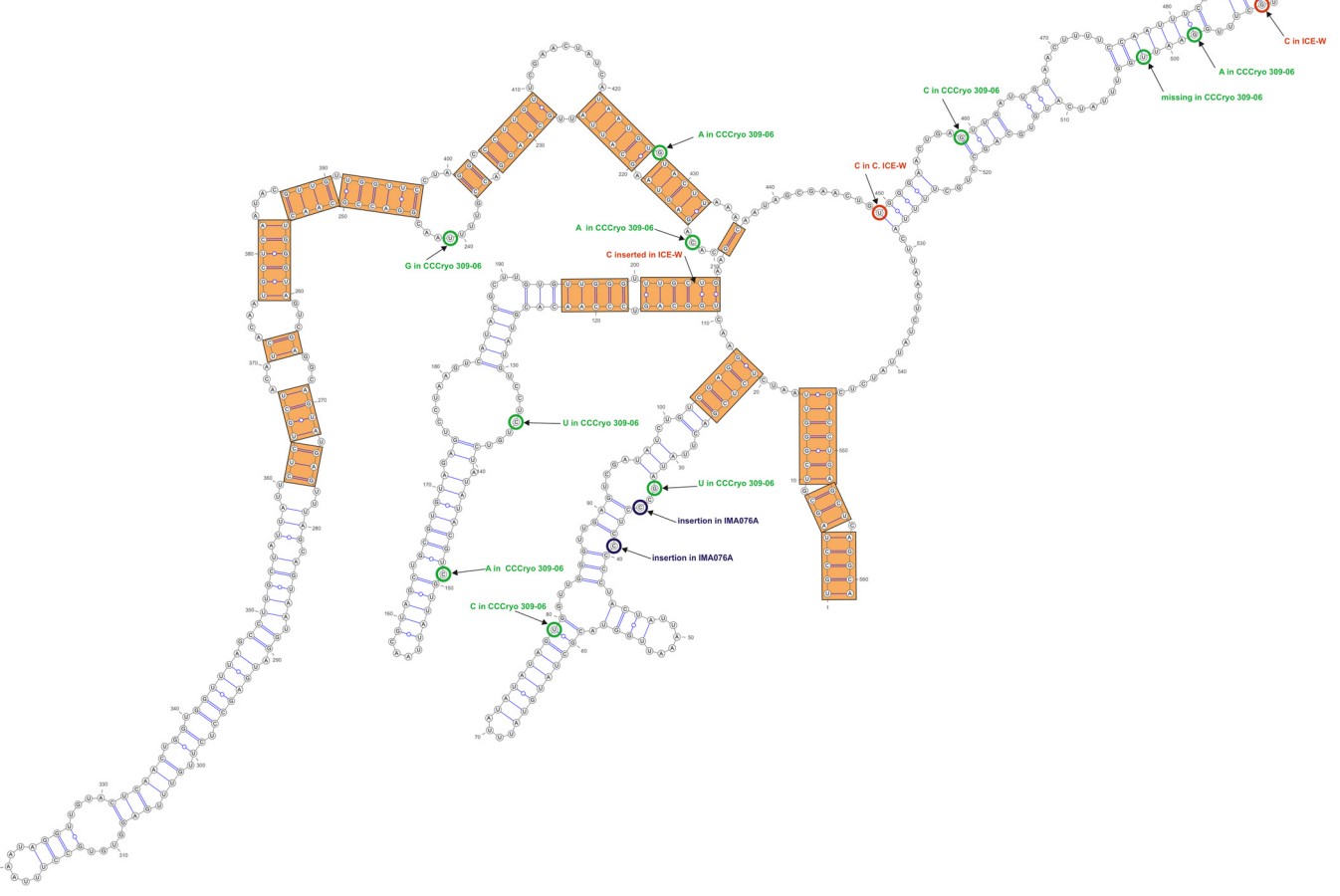

**Figure 3.** Hypothetical secondary structure of the ITS-2 spacer for *Microglena antarctica* sp. nov. The nucleotide sequences of the spacers between the four main helices are reported on the structure. A-U pairings are represented with a single line, G-C pairings with a double line, and unconventional pairings with a line interrupted by a circle. Orange boxes include the barcode regions. Nucleotide differences between *M. antarctica* IMA076A and other strains of the same species are represented by coloured circles (light green for strain CCCryo 309-06, red for strain ICE-W and blue for insertion detected only in IMA076A).

**Table 2.** ITS-2 DNA Barcode of the *Microglena* species. Base-pair alignment is coded using a number for each base-pair (1 = A-U, 2 = U-A, 3 = G-C, 4 = C-G, 5 = G•U, 6 = U•G, 7 = mismatch and 8 = deletion, unpaired of single bases). ★ = *M. opisthopyren* [12], ● = *M. media* [12]. * = non-homoplasious synapomorphies. ** = 2 non-homoplasious synapomorphies.

| Species - Strains | Barcode | 5.8S-LSU | Helix I | Helix II | Helix III |
|---|---|---|---|---|---|
| *M. antarctica* IMA076/CCCryo 309-06/ICE-L/ICE-W | A | 234421342453326 | 64243 | 65341574441 | 3888885132188541822154113388448883514454114235361311313613138 |
| | | ............... | ..*.. | ........... | ..................**.........................*...*. |
| *M. redcarensis* SAG 18. 89 | B | 234421342453326 | 64141 | 65341574441 | 5888183136881348821254113518444888851445411423536131134130148 |
| | | ............... | ....* | ........... | *.........*........**......*.......*............... |
| *M. reginae* SAG 17.89 | C | 234421342453327 | 64144 | 63344774441 | 3382183158888181848532113318442133554454114855388588321384 |
| | | ...............* | ..... | .*...*..... | .*...................*..........**..........*.........*..... |
| *M. uva-maris* SAG 19.89 | D | 234421342453326 | 64143 | 65141374441 | 32811831588881828483348133184425135544541142353215885413848 |
| | | ............... | ..... | ..*........ | ...*...........*..........**..........*...... |
| *M. lobata* SAG 31.72 | E | 234421342453326 | 64124 | 65347374441 | 38821831338885838483888133844488833344541186535215883413848 |
| | | ............... | ..... | ........... | ........................................................... |
| *M. braunii* SAG 50.86 | F | 234421342453326 | 64112 | 65342374441 | 38821831331885838483888531444888333445411865352158883413848 |
| | | ............... | ...*. | ........... | ...........................*............................... |
| *M. globulifera* CCAC 0015/CCAC 0017 | G | 234421342453326 | 64126 | 65344374641 | 36828831538881818485888133844488833344541162553881883413848 |
| | | ............... | ....* | ........*.. | .....................................*..................... |
| *M. longirubra* SAG 5.92 | H | 234421342453326 | 64142 | 65344374441 | 36828851338813818483888133844488835344541126553285883413848 |
| | | ............... | ..... | ........... | ........................................................... |
| *M. coccifera* SAG 54.91/SAG 55.91 | I | 234421342453326 | 64142 | 65344374441 | 36821831338815878787887133844488833344541122553885883413848 |
| | | ............... | ..... | ........... | ................*.*.*..* ................................. |
| *M. opisthopyren* SAG 8.87/SAG 54.90/CCAP 11/46 ★ | J1 | 234421342453326 | 64142 | 65344374441 | 36821831338883838483888133844488833344541122553885883413848 |
| | | ............... | ..... | ........... | ........................................................... |
| *M. opisthopyren* NIES-2744 ★ | J2 | 234421342453326 | 64142 | 65344374441 | 32821831338883858483888133844488833344541122553885883413846 |
| | | ............... | ..... | ........... | ..........................................................* |
| *M. basinucleata* SAG 67.72 | K | 237421342453326 | 64142 | 65342374441 | 32831833328681818483841833884488855344541142535885883413848 |
| | | ..*............ | ..... | ........... | ...........*....................*.......................... |
| *M. monadina* SAG 55.72 | L | 234421342453326 | 64142 | 65344374441 | 38888331888838884838888833844488833344541142535215883413848 |
| | | ............... | ..... | ........... | ......*..*................................................ |
| *M. charkoviensis* ACKU267-03/ACKU274-03 | M | 234421342453326 | 64144 | 65344374441 | 36838813338265851483848833844488833344541142553881883413848 |
| | | ............... | ..... | ........... | ..........**...*.......................................... |
| *M. indica* SAG 46.96 | N | 234421342453326 | 64124 | 65363374441 | 32821833318883858485888833844488833344541142553685883413848 |
| | | ............... | ..... | ...**...... | ........................................................... |
| *M. media* SAG 16.90 ● | O1 | 234421342453326 | 64164 | 65347374441 | 38861833318883858485841833184488833344541122553685883413842 |
| | | ............... | ..... | ........... | ...*...................................................... |
| *M. media* NIES-2743 ● | O2 | 234421342453326 | 64164 | 65347374441 | 32121833318883858485841833184488833344541122553685883413842 |
| | | ............... | ..... | ........... | ..*....................................................... |

### 3.4. Cell Morphology and Ultrastructure

Through differential interference contrast, scanning electron and confocal microscope observations, strain IMA076A appeared as a unicellular motile green alga, characterized by the presence of two flagella (Figure 5a–c), projecting from the apical part of the cell. Cells were oval in shape with average length and width of $17 \pm 3$ μm and $12 \pm 3$ μm, respectively (Figure 5a,b,d). Moreover, the cells were characterized by the presence of bulges, emerging from the cell surface (Figure 5b). Asexual reproduction was by sporulation with the production of two or four zoospores (Figure 5c). TEM investigations confirmed the oval shape of the cells, with the nucleus located in the anterior-central position (Figure 5d). A single cup-shaped chloroplast with a thick basal part, occupied almost all the cell volume (Figure 5d) and was characterized by a pyrenoid. This was spherical or ellipsoidal in shape, surrounded by a single layer of several starch platelets (Figure 5d,f) and penetrated by thylakoid membranes (Figure 5f). A trapezoidal wall papilla with the insertion of flagella and contractile vacuoles were located in the apical part of the cell (Figure 5e). Morpho-ecological data of IMA076A and phylogenetically related species (Table 3) were visualized in Figure 6.

**Table 3.** Morphological characters and ecology of *Microglena* species.

| Species | Chloroplast Morphotype | Pyrenoid-Shape | Pyrenoid Fragmentation | Pyrenoid-Starch Granules | Basal Thickening of the Chloroplast of Mature Cells | Nucleus-Position | Cell Shape | Ecology |
|---|---|---|---|---|---|---|---|---|
| *M. uva-maris* | (I) | ellipsoid | absent | several and large | present | central | ellipsoid to widely ellipsoidal | marine |
| *M. redcarensis* | (I) | ellipsoid | absent | several and large | present | central | ellipsoid to spherical | marine |
| *M. reginae* | (I) | ellipsoid | absent | many and small | present | central | ellipsoid to oval to spherical | marine |
| *M. antarctica* | (I) | ellipsoid to spherical | absent | several and large | present | central-anterior | ellipsoid to oval | marine |
| *M. globulifera* | (III) | halfring | present | many and small | absent | central | ellipsoidal to spherical | freshwater |
| *M. braunii* | (II) | halfring | sometimes fragmented | many and small | absent | central | ellipsoidal to widely ellipsoidal to spherical | freshwater |
| *M. lobata* | (II) | halfring | absent | many and small | absent | central | ellipsoid to widely ellipsoidal | freshwater |
| *M. longirubra* | (II) | halfring | absent | many and small | absent | central | ellipsoid to widely ellipsoidal | freshwater |
| *M. charkoviensis* | (I) | ellipsoid | absent | many and small | absent | central | ellipsoid to widely ellipsoidal | freshwater |
| *M. monadina* | (II) | halfring | absent | many and small | absent | central | ellipsoid to widely ellipsoid | freshwater |
| *M. basinucleata* | (II) | halfring | absent | no starch | absent | basal | ellipsoid to widely ellipsoid | freshwater |
| *M. coccifera* | (III) | ellipsoid to spherical | present | many and small | present | central | widely ellipsoid | freshwater |
| *M. indica* | (II) | halfring | absent | many and small | absent | central | widely ellipsoid to spherical | freshwater |
| *M. opisthopyren* | (II) | halfring to ellipsoid | sometimes fragmented | many and small | present | central | ellipsoid to widely ellipsoid | freshwater |
| *M media* | (II) | ellipsoid | sometimes fragmented | many and small | absent | central | ellipsoid to widely ellipsoid | freshwater |

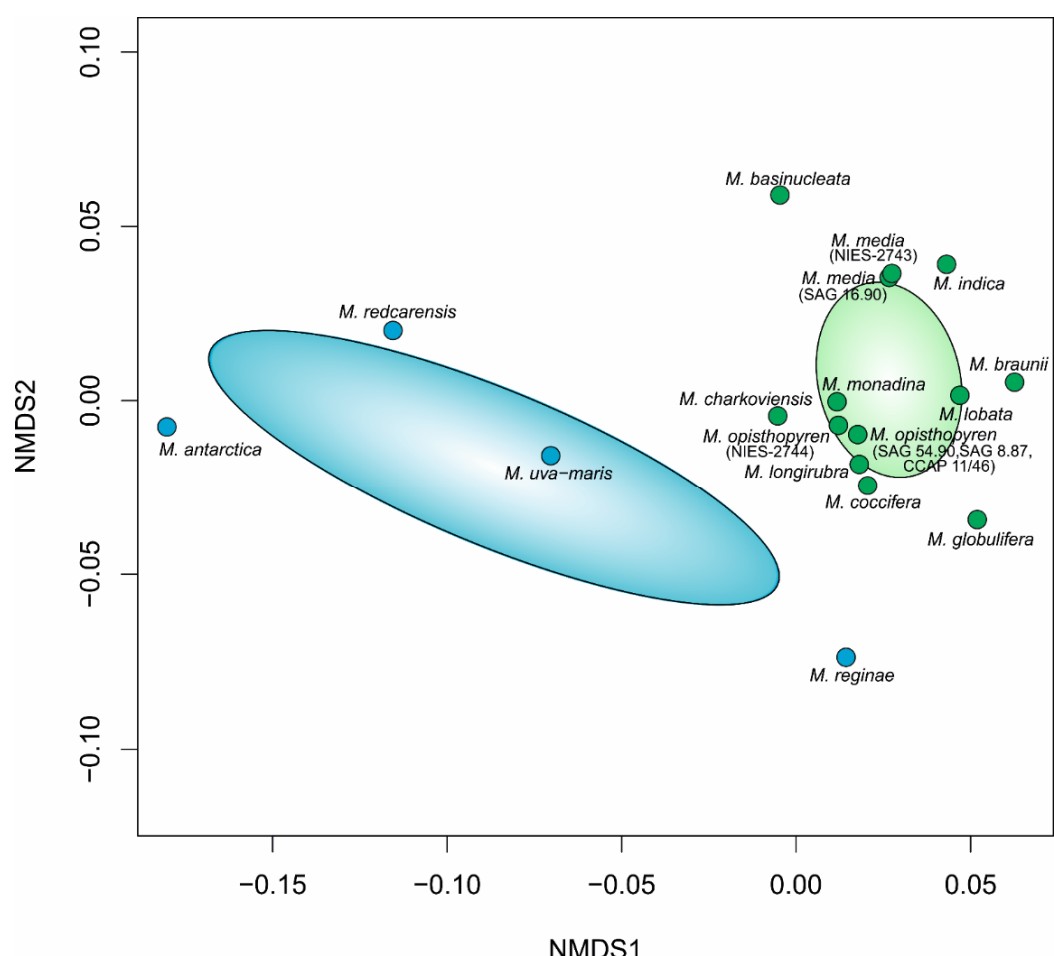

**Figure 4.** Non-metric multidimensional scaling of p-distances of marine (blue) and freshwater (green) *Microglena* species.

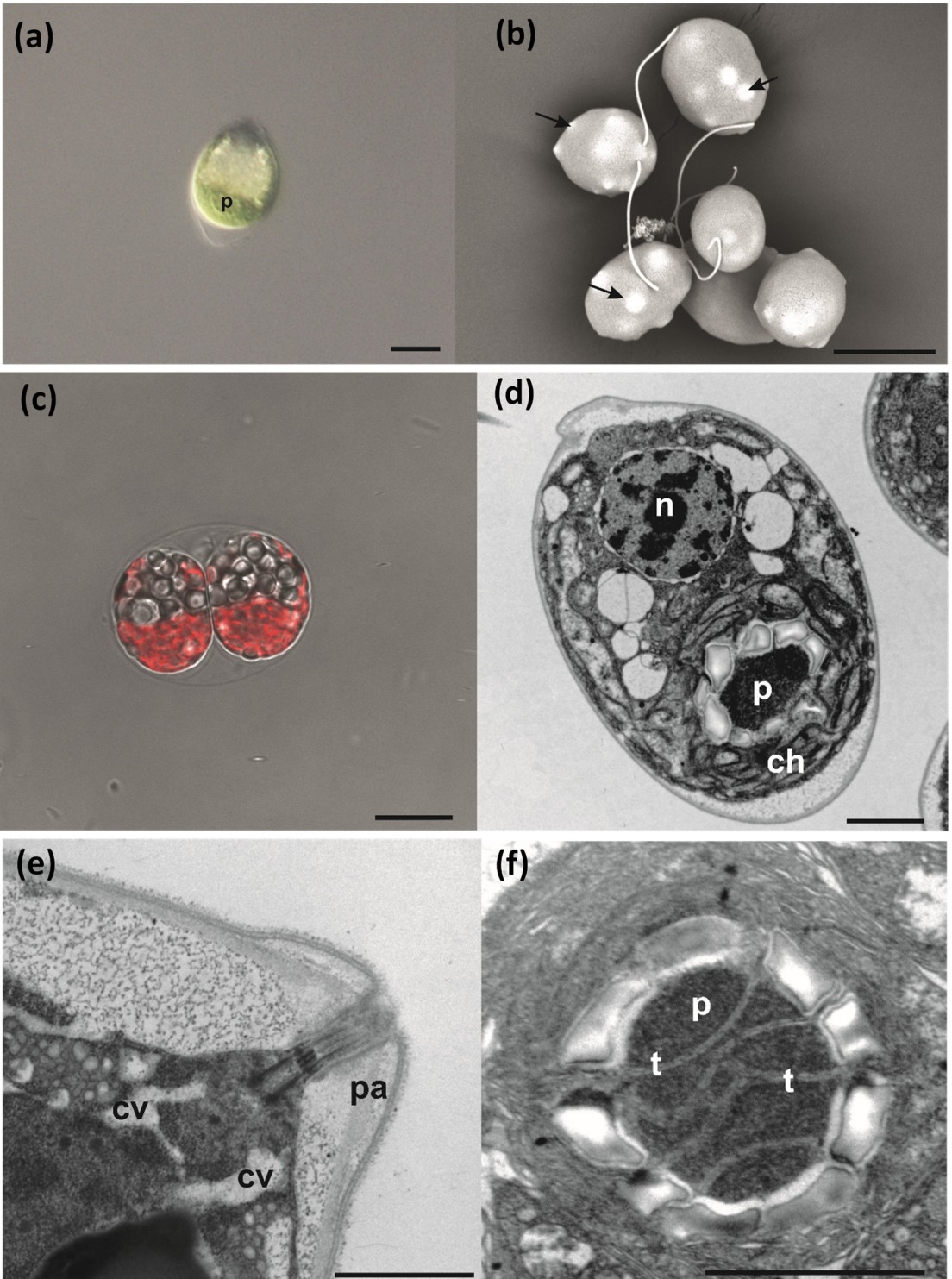

**Figure 5.** Differential interference contrast (**a**), scanning electron (**b**), confocal (**c**) and transmission electron (**d**–**f**) microscopy images of *Microglena antarctica* sp. nov. Single cell showing the cup-shaped chloroplast characterized by the presence of a pyrenoid (p) (**a**), overview of vegetative cells showing the presence of bulges emerging from the cell surface (arrows) (**b**), asexual reproduction by sporulation with production of zoospores (**c**), longitudinal section of a vegetative cell showing a cup-shaped chloroplast (ch) with a pyrenoid (p) and a nucleus in anterior-central position (n) (**d**), detail of the apical part of the cell showing the papilla (pa) and contractile vacuoles (cv) (**e**), detail of pyrenoid (p) showing thylakoid membranes (t). Scale bars: 10 μm (**a**), 10 μm (**b**), 10 μm (**c**), 2 μm (**d**), 1 μm (**e**), 2 μm (**f**).

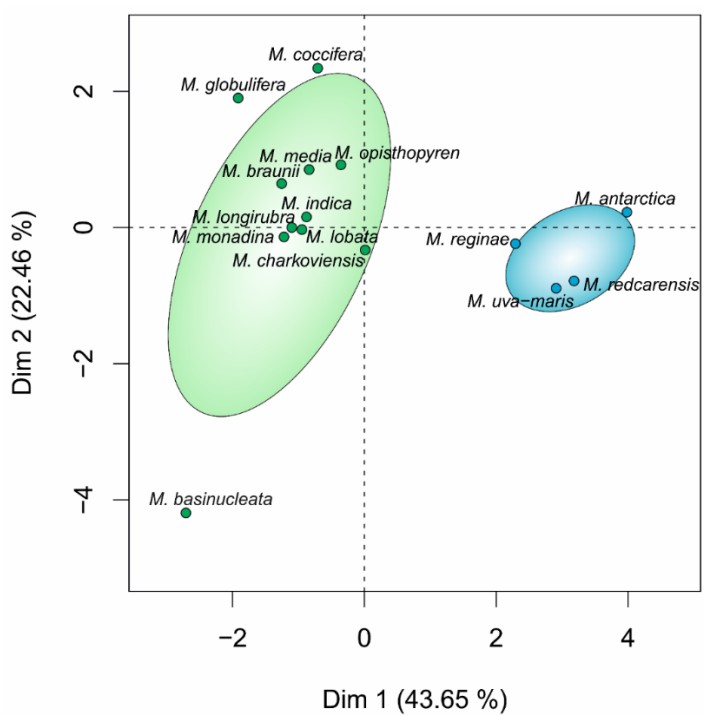

**Figure 6.** Principal component map showing morphological and ecological dissimilarities between marine (blue) and freshwater (green) *Microglena* species.

### *3.5. Taxonomy*

Our findings strongly support the view that the Antarctic strains belong to a new species deserving the inclusion in the genus *Microglena*.

*Microglena antarctica* Trentin, Negrisolo, Moschin, Veronese, Cecchetto & I.Moro sp. nov.

***DESCRIPTION:*** Ovoid vegetative cells with two flagella, 11.9–24.4 μm in length and 7.1–19.1 μm in diameter, with a granular cell surface. Chloroplast cup-shaped with a thick basal part. Pyrenoid spherical to ellipsoidal, surrounded by a single layer of several starch platelets. Lateral bright stigma composed of small drops arranged in two layers, located in medial position. Nucleus medial-anterior. Asexual reproduction by sporulation into two or four zoospores. Sexual reproduction not observed.

***HOLOTYPE:*** Strain IMA076A resin-embedded sample deposited in the Italian National Antarctic Museum (MNAIT, Section of Genoa) with the voucher code MNA147301.

***TYPE LOCALITY:*** Inexpressible Island (Penguin Lagoon, Terra Nova Bay, Ross Sea, Antarctica), coordinates: 74°54′ S 163°39′ E

***ETYMOLOGY:*** The species name 'antarctica' is derived from the continent where it was collected.

***DNA SEQUENCE AVAILABLE:*** ON185622 (18S rDNA) and OM791388 (ITS-2).

### 4. Discussion

Since green unicellular algae, especially *Chlamydomonas*-like species, have similar cell morphologies [4], we mainly relied on the use of molecular data for the taxonomic identification of the Antarctic isolate IMA076A. Our phylogenetic analysis showed that the Antarctic strains belong to a new species *Microglena antarctica*. This taxon was not closely related to any well-established species, as shown by the molecular phylogenetic analysis, CBCs and p-distances comparisons, as well as the application of five species delimitation methods. In our phylogenetic reconstruction, the Antarctic strains (IMA076A, ICE-L, ICE-W and CCCryo 309-06) grouped with *M. redcarensis* SAG 18.89 and formed a well-supported clade (100% SH-aLRT value, 1 posterior probability, 100% bootstrap). Our best tree resolved three lineages within the genus *Microglena*: 'clade I' (formed by *M. antarctica* IMA076A, CCCryo 309-06, ICE-L and ICE-W and *M. redcarensis* SAG 18.89)

and 'clade II' (including *M. reginae* SAG 17.89 and *M. uva-maris* SAG 19.89), consisting of marine species, and 'clade III', consisting of freshwater species. We tested different tree topologies to evaluate alternative hypotheses of the evolution of the genus *Microglena*. A tree constrained to monophyly all the marine *Microglena* species was not significantly different from our best unconstrained tree. However, the recoveries of 'clade I' together with 'clade III' and of *M. redcarensis* SAG 18.89 with 'clade II' were significantly worse than the best unconstrained tree.

The evidence gathered in recent investigations on the biflagellate unicellular green alga suggests that closely related groups to the genus *Microglena*, such as *Moewusinia* clade [51], ancestrally inhabits freshwater, while other algae in *Xenovolvoxa*, such as clade *Phacotinia*, and *Chlamydomonas chlamydogama* could only be found in freshwater habitats [11,52–54]. As proposed by different authors, *Microglena* species likely evolved from freshwater ancestry [4,11]. Despite the monophyly of *Microglena* is supported by several DNA-based phylogenies [4,11,12], including the one reported here, none of these studies focused on the evolution of the marine subclades within *Microglena*.

*M. antarctica* (IMA076A, CCCryo 309-06, ICE-L and ICE-W), *M. redcarensis* SAG 18.89, *Microglena reginae* SAG 17.89 and *M. uva-maris* SAG 19.89 could represent the transition state from freshwater to marine and polar environments. Particularly, *Microglena antarctica* tolerates a wide salinity range [55], a possible adaptation to Antarctic environmental conditions, where salinity could vary during the freezing and melting process of water. The growth rate of *Microglena antarctica* strain ICE-L has been reported to decrease with the increasing salt concentration [55], indicating that it may prosper during the summer when the ice pack melts. This hypothesis is corroborated by the characteristics of the sampling area where *Microglena antarctica* IMA076A was collected; specifically, the Penguin Lagoon is a transitional environment with high salinity fluctuations. Another possible adaptation to marine life is represented by the activity of contractile vacuoles. Freshwater *Microglena* species have two anterior contractile vacuoles, which can be also observed in *Microglena uva-maris* SAG 19.89, if cultivated in freshwater (3N-BBM þ V) medium [4]. *Microglena reginae* SAG 17.89 is unable to grow in freshwater conditions and its contractile vacuoles do not function in seawater [4], while *Microglena redcarensis* SAG 18.89 activates its contractile vacuoles in freshwater as a short-term defence response against low osmotic pressure [11]. *Microglena antarctica* showed two anterior contractile vacuoles if cultivated in salt water. Future investigations on the activity of contractile vacuoles may shed light on the role of these organelles in the adaptation in freshwater and in marine environments. Furthermore, marine species are characterized by the presence of a cup-shaped chloroplast with a thick basal part, in which a single pyrenoid is located. The morphological features of this shape of chloroplast were described by Demchenko et al., 2012 [4] and referred to as morphotype I. However, this morphotype is not a unique characteristic of the marine species, and could be also detected in *M. charkoviensis* (ACKU 267-03 and ACKU 274-03). Despite *M. charkoviensis* being isolated from a freshwater habitat, future studies could assess its physiological responses to different ranges of salinities. This could shed light on the variability of chloroplast morphotypes within this genus and on the evolution of this organelle.

## 5. Conclusions

The present work upholds the importance of sequence-based approaches in the recognition of new species and genera. The discovery of a new species of Chlorophyceae in Antarctica highlights that the photosynthetic diversity of the Ross Sea is more extensive than previously thought. A current problem for reliable molecular analysis of psychrophilic algae is considered to be the low resolution of the 18S rDNA marker at the species level [5]. In this sense, more strains will probably have to be transferred to the genus *Microglena*. This could be the case for some Antarctic strains described through the 18S rDNA marker and for which complete ITS-2 sequences are not yet available. Another issue in the resolution of the molecular phylogeny of the '*Microglena* clade' is the lack of sequences of other molecular

markers, such as *rbc*L and *tuf*A. For the detection of these cryptic species, more variable barcodes might be used to unambiguously identify different taxa, bearing in mind that a single barcode represents only a fraction of an organism's variation [56].

**Supplementary Materials:** The following supporting information can be downloaded at: https://www.mdpi.com/article/10.3390/d14050337/s1. Table S1. List of species, strains and GenBank accession numbers of sequences used in this study. Table S2. Compensatory base changes (CBCs) and uncorrected p-distances among the ITS-2 DNA barcodes of Microglena. The upper-right half of the table shows the total number of compensatory changes (CBC/Hemi-CBCs), while the lower-left half gives the uncorrected p-distances calculated in PAUP. Figure S1. 18S rDNA ML tree. Approximate Likelihood Ratio Tests based on Shimodaira-Hasegawa-like procedures (SH-aLRT) values (%), Bayesian Posterior Probabilities (PP) and ML bootstrap values (%) are shown above branches. Figure S2. ITS-2 ML tree. Approximate Likelihood Ratio Tests based on Shimodaira-Hasegawa-like procedures (SH-aLRT) values (%), Bayesian Posterior Probabilities (PP) and ML bootstrap values (%) are shown above branches.

**Author Contributions:** Conceptualization, R.T., E.N. and I.M.; methodology, R.T., E.N.; software, R.T. and D.V.; validation, R.T., E.N. and I.M.; formal analysis, R.T., E.N., E.M. and D.V.; investigation, R.T., E.N., E.M., M.C., D.V. and I.M.; resources, I.M.; data curation, R.T. and D.V.; writing—original draft preparation, R.T. and D.V.; writing—review and editing, R.T., E.N., E.M., M.C., D.V. and I.M.; visualization, R.T.; supervision, E.N. and I.M.; project administration, I.M.; funding acquisition, I.M. All authors have read and agreed to the published version of the manuscript.

**Funding:** This research was funded by the PNRA project PNRA 16 00120–TNB-code: 'Terra Nova Bay barcoding and metabarcoding of Antarctic organisms from marine and limno-terrestrial environments' (PI: S. Schiaparelli). Authors are grateful to the Italian National Antarctic Scientific Commission (CSNA) and the Italian National Antarctic program (PNRA) for the endorsement of the Special Issue initiative and to the Italian National Antarctic Museum (MNA) for the financial support.

**Institutional Review Board Statement:** Not applicable.

**Data Availability Statement:** Not applicable.

**Acknowledgments:** We wish to thank the Electron Microscopy Facility of the Department of Biology at the University of Padova.

**Conflicts of Interest:** The authors declare no conflict of interest.

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
