# Peer review of "Microglena antarctica sp. nov. a New Antarctic Green Alga from Inexpressible Island (Terra Nova Bay, Ross Sea) Revealed through an Integrative Approach"

_diversity, doi:10.3390/d14050337_

Round 1
Reviewer 1 Report
Page 2, lines 68-71: I don’t think it is necessary to give the authority of the genus and species at this point. You will state the authority in the formal descriptions.
Page 2, line 69: “that differs from the genus Microglena” – how does it differ?
Page 2, line 72: this finding or these findings?
Page 4, line 177: what does it mean OLD ‘Monadina clade’? Also, consider using Monadinia (in italics)
Page 6, Figure 2: please, include the explanation of the colored columns in the legend.
Page 9, Table 1: It is very confusing to understand this table. It is not reader friendly at all. Could you please organize it better? Alternatively, you could split the data and move CBSs or p-distances to the Supplementary material.
Page 10, line 278: “apical side” – maybe apical part?
Page 10, line 279: diameter or width?
Page 10, line 281: what does it mean by cell division? Did the alga form zoospores or aplanospores?
Page 10, line 282: “distinction of an apical side” – what do you mean here?
Page 11, line 283: nucleus enclosed in the chloroplast??? Are you stating that nucleus is in the chloroplast? This makes no sense.
Page 11, line 289: “apical side” – apical part
Page 11, line 296: I don’t see any flagellar apparatus here. Just a smooth cell wall, two flagella, and a papilla.
Page 12, Figure 7: it would be nice if you could mark the discussed structures (either by letters or arrows/arrowheads) in the microphotographs.
Page 13, Table 2: Please, give the explanation what does chloroplast type I, II, III mean.
Page 15, The International Code of Nomenclature for algae, fungi and plants say: 38.1. In order to be validly published, a name of a new taxon (see Art. 6.9) must (a) be accompanied by a description or diagnosis of the taxon (see also Art. 38.7 and 38.8). Not both! So, please choose either description or diagnosis.
Also, the initials of author names should be use only if another person with the same surname exists. For example, you can check it here: https://www.ipni.org/
Page 15, 320: papilla does not look like crown to me…
Page 16, line 378: “in” written two times
Page 16, line 412: specially or especially?
Page 16, line 413: rely or relied?
Page 16, line 414: shows or showed? Generally, the results are written in the past tense.
Page 17, line 420: I would say it formed moderately supported sister clade. In fact, based on the support values (75/0.99/89) I would not conclude that M. reginae and M. uva-maris belong to the genus Chloroplanis. However, I understand the logic here and your point of view. Maybe you could explicitly state that the transfer of these two species into genus Chloroplanis is also supported by morphological and ecological data.
Page 17, Moewusinia should be in italics
Page 17, line 433: Phacotinia should be in italics
Page 17, line 458: “more extensive than previously thought” – but what was previously thought?
Page 17, line 459: is Chloroplanis antarctica really a psychrophilic alga? Did you test it?
Page 17, line 462: could you please compute the extensive 18S tree showing all close relatives of Chloroplanis and most importantly its position within the Monadinia, and upload it as a supplementary file?
The authors sate they sequenced partial rbcL and tufA genes. However, it is unclear what did they do with these data? No results or discussion follow up. Why?
References 7 and 18 are in capital letters.
Author Response
Dear reviewer 1,
We are grateful to your observations and comments. We have made new phylogenetic analyses based on the full-length 18S rDNA sequences, generated for our strain IMA076A and retrived in the Genbank achives for the other species analysized, and on the ITS-2 sequences. We provided the concatenated ML tree in the text of the manuscript and the single marker trees (18S rDNA and ITS-2) in the supplementary material. Furthermore, we evaluated alternative topologies in IQTree for our concatenated tree. The results are reported in the manuscript (Table 1).
In short, in light of the results of our new analysies, we considered the genus Microglena as divided in three subclades (two marine caldes and one freshwater lineage). ‘Clade I’ included M. antarctica and M. redcarensis, ‘clade II’ included M. reginae and M. uva-maris and ‘clade III’ included freshwater species. The monophyletic origin of the marine Microglena species is supported topologically, but further analyses are required to establish if this group correspond to a new genus or not (for example, increasing taxon sequencing may affect these results).
In conclusion, we believe that our molecular results support the identification of a new Antarctic species within the genus Microglena. We describe this specie as Microglena antarctica sp. nov., whereas alternate hypotheses on the evolution of marine and Antarctic species should be explored in the future.
Here are our point-to-point responses to your comments:
1) Page 2, lines 68-71: I don’t think it is necessary to give the authority of the genus and species at this point. You will state the authority in the formal descriptions.
RESPONSE 1) There is no general rule about this. Despite it is not strictly necessary, we decided to keep this sentence to help the reader.
2) Page 2, line 69: “that differs from the genus Microglena” – how does it differ?
RESPONSE 2) We deleted this sentence.
3) Page 2, line 72: this finding or these findings?
RESPONSE 3) Recommendation accepted.
4) Page 4, line 177: what does it mean OLD ‘Monadina clade’? Also, consider using Monadinia (in italics)
RESPONSE 4) We modified the text. Recommendation accepted.
5) Page 6, Figure 2: please, include the explanation of the colored columns in the legend.
RESPONSE 5) We modified the legend, according to your indications.
6) Page 9, Table 1: It is very confusing to understand this table. It is not reader friendly at all. Could you please organize it better? Alternatively, you could split the data and move CBSs or p-distances to the Supplementary material.
RESPONSE 6) Recommendation accepted. We moved the table to the supplementaty material section.
7) Page 10, line 278: “apical side” – maybe apical part?
RESPONSE 7) Recommendation accepted.
8) Page 10, line 279: diameter or width?
RESPONSE 8) Recommendation accepted.
9) Page 10, line 281: what does it mean by cell division? Did the alga form zoospores or aplanospores?
RESPONSE 9) We have changed in the text the sentence about the reproduction.
10) Page 10, line 282: “distinction of an apical side” – what do you mean here?
RESPONSE 10) We modified the sentence.
11) Page 11, line 283: nucleus enclosed in the chloroplast??? Are you stating that nucleus is in the chloroplast? This makes no sense.
RESPONSE 11) We modified the text.
12) Page 11, line 289: “apical side” – apical part
RESPONSE 12) Recommendation accepted.
13) Page 11, line 296: I don’t see any flagellar apparatus here. Just a smooth cell wall, two flagella, and a papilla.
RESPONSE 13) We modified the sentence
14) Page 12, Figure 7: it would be nice if you could mark the discussed structures (either by letters or arrows/arrowheads) in the microphotographs.
RESPONSE 14) We added letters and arrows in the micrographs.
15) Page 13, Table 2: Please, give the explanation what does chloroplast type I, II, III mean.
RESPONSE 15) Different morphotypes are described in Demchenko et al., 2012. We cited this study in the text when referring to the chloroplast morphotype.
16) Page 15, The International Code of Nomenclature for algae, fungi and plants say: 38.1. In order to be validly published, a name of a new taxon (see Art. 6.9) must (a) be accompanied by a description or diagnosis of the taxon (see also Art. 38.7 and 38.8). Not both! So, please choose either description or diagnosis.
RESPONSE 16) Thanks for the suggestion. We chose to put only the description of the species.
17) Also, the initials of author names should be use only if another person with the same surname exists. For example, you can check it here: https://www.ipni.org/
RESPONSE 17) Recommendation accepted.
18) Page 15, 320: papilla does not look like crown to me…
RESPONSE 18) We modified the sentence.
19) Page 16, line 378: “in” written two times
RESPONSE 19) Recommendation accepted.
20) Page 16, line 412: specially or especially?
RESPONSE 20) Recommendation accepted.
21) Page 16, line 413: rely or relied?
RESPONSE 21) Recommendation accepted.
22) Page 16, line 414: shows or showed? Generally, the results are written in the past tense.
RESPONSE 22) Recommendation accepted.
23) Page 17, line 420: I would say it formed moderately supported sister clade. In fact, based on the support values (75/0.99/89) I would not conclude that M. reginae and M. uva-maris belong to the genus Chloroplanis. However, I understand the logic here and your point of view. Maybe you could explicitly state that the transfer of these two species into genus Chloroplanis is also supported by morphological and ecological data.
RESPONSE 23) As indicated at the beginning, we now describe IMA076A as a new species of Microglena, to avoid this problem.
24) Page 17, Moewusinia should be in italics
RESPONSE 24) Recommendation accepted.
25) Page 17, line 433: Phacotinia should be in italics
RESPONSE 25) Recommendation accepted.
26) Page 17, line 458: “more extensive than previously thought” – but what was previously thought?
RESPONSE 26) We deleted this sentence
27) Page 17, line 459: is Chloroplanis antarctica really a psychrophilic alga? Did you test it?
RESPONSE 27) It is reported in many studies. See:
-An, Meiling; Mou, Shanli; Zhang, Xiaowen; Ye, Naihao; Zheng, Zhou; Cao, Shaona; Xu, Dong; Fan, Xiao; Wang, Yitao; Miao, Jinlai (2013). Temperature regulates fatty acid desaturases at a transcriptional level and modulates the fatty acid profile in the Antarctic microalga Chlamydomonas sp. ICE-L. Bioresource Technology, 134(), 151–157. doi:10.1016/j.biortech.2013.01.142
-An, Meiling; Mou, Shanli; Zhang, Xiaowen; Zheng, Zhou; Ye, Naihao; Wang, Dongsheng; Zhang, Wei; Miao, Jinlai (2013). Expression of fatty acid desaturase genes and fatty acid accumulation in Chlamydomonas sp. ICE-L under salt stress. Bioresource Technology, 149(), 77–83. doi:10.1016/j.biortech.2013.09.027
-Zhang, Zhenhua; An, Meiling; Miao, Jinlai; Gu, Zhiqiang; Liu, Chang; Zhong, Bojian (2018). The Antarctic sea ice alga Chlamydomonas sp. ICE-L provides insights into adaptive patterns of chloroplast evolution. BMC Plant Biology, 18(1), 53–. doi:10.1186/s12870-018-1273-x
28) Page 17, line 462: could you please compute the extensive 18S tree showing all close relatives of Chloroplanis and most importantly its position within the Monadinia, and upload it as a supplementary file?
RESPONSE 28) Thanks for the suggestion. We uploaded as a new supplementary file.
29) The authors sate they sequenced partial rbcL and tufA genes. However, it is unclear what did they do with these data? No results or discussion follow up. Why?
RESPONSE 29) We deleted this part in the text as suggested by other reviewers.
30) References 7 and 18 are in capital letters.
RESPONSE 30) Recommendation accepted.
Reviewer 2 Report
This paper aims to explore the diversity of microalgae in Antarctic region, and recognize the new genus of green algae based on combined morphological and molecular approaches. I think this work provide insight into diversity of Antarctic microalgae as well as Chlorophyceae. I have few comments and suggestion, please see below.
- Abstract: if possible, I think author should shorten this statement "Combining: 1) phylogenetic analyses of the V4 region of the small subunit rDNA (18S rDNA) and of the internal transcribed spacer 2 (ITS-2) sequences; 2) species delimitation methods; 3) comparative analyses of the secondary structures of ITS-2 and compensatory base changes; 4) morphological, ultrastructural and ecological features", and provide a bit more information about how this new genus differs morphologically and ecologically from other closely related genera. Also mention about the phylogenetic position of this genus in Chlorophyceae. What is phylogenetically related genera ?
- Also authors mentioned "compensatory base changes (CBCs) approach are pivotal in the identification of cryptic microalgae", and I did not see how this help to clarified cryptic species. I think probably species delimitation may help more clear than CBCs.
- Figure 2. Please label new sequences generated in this present study in bold, and put the braces to indicate the clade of new genus.
- Could authors make Table 1 as supplementary document?
Author Response
Dear reviewer 2,
We are grateful to your observations and comments. We have made new phylogenetic analyses based on the full-length 18S rDNA sequences, generated for our strain IMA076A and retrived in the Genbank achives for the other species analysized, and on the ITS-2 sequences. We provided the concatenated ML tree in the text of the manuscript and the single marker trees (18S rDNA and ITS-2) in the supplementary material. Furthermore, we evaluated alternative topologies in IQTree for our concatenated tree. The results are reported in the manuscript (Table 1).
In short, in light of the results of our new analysies, we considered the genus Microglena as divided in three subclades (two marine caldes and one freshwater lineage). ‘Clade I’ included M. antarctica and M. redcarensis, ‘clade II’ included M. reginae and M. uva-maris and ‘clade III’ included freshwater species. The monophyletic origin of the marine Microglena species is supported topologically, but further analyses are required to establish if this group correspond to a new genus or not (for example, increasing taxon sequencing may affect these results).
In conclusion, we believe that our molecular results support the identification of a new Antarctic species within the genus Microglena. We describe this specie as Microglena antarctica sp. nov., whereas alternate hypotheses on the evolution of marine and Antarctic species should be explored in the future.
Here are our point-to-point responses to your comments:
‘This paper aims to explore the diversity of microalgae in Antarctic region, and recognize the new genus of green algae based on combined morphological and molecular approaches. I think this work provide insight into diversity of Antarctic microalgae as well as Chlorophyceae. I have few comments and suggestion, please see below’.
1) Abstract: if possible, I think author should shorten this statement "Combining: 1) phylogenetic analyses of the V4 region of the small subunit rDNA (18S rDNA) and of the internal transcribed spacer 2 (ITS-2) sequences; 2) species delimitation methods; 3) comparative analyses of the secondary structures of ITS-2 and compensatory base changes; 4) morphological, ultrastructural and ecological features", and provide a bit more information about how this new genus differs morphologically and ecologically from other closely related genera. Also mention about the phylogenetic position of this genus in Chlorophyceae. What is phylogenetically related genera ?
RESPONSE 1) We modified the text. Recommendation accepted.
2) Also authors mentioned "compensatory base changes (CBCs) approach are pivotal in the identification of cryptic microalgae", and I did not see how this help to clarified cryptic species. I think probably species delimitation may help more clear than CBCs.
RESPONSE 2) Ok, thanks for the suggestion. We performed both analyses. As reported by Coleman 2000
and Demchenko et al. 2012, the identification of cryptic Chlorophyceae mainly rely on the CBCs approach.
3) Figure 2. Please label new sequences generated in this present study in bold, and put the braces to indicate the clade of new genus.
RESPONSE 3) We modified the figure as indicated.
4) Could authors make Table 1 as supplementary document?
RESPONSE 4) Recommendation accepted. We moved the tabele to the supplementaty material section.
Reviewer 3 Report
The manuscript under review presents new genus of green algae revealed in green-ice sample collected at Inexpressible Island (Antarctica). Authors isolated this alga in the culture and studied it using molecular phylogenetic methods, light, confocal, scanning electron and transmission electron microscopy methods. Based on the molecular data, phenotypic features observed in the Antarctic alga and ecology (marine habitat) authors conclude that it represents a new genus described as Chloroplanis gen. nov. Beside Chloroplanis antarctica (newly described type species) they transfer to the new genus three species, formerly members of the genus Microglena (Chloroplanis uva-maris, Chloroplanis reginae, and Chloroplanis redcarensis) and showed that a number of strains in the culture collections also represent C. antarctica.
In my opinion C. antarctica is a new species indeed and deserves description. However, entity of the new genus Chloroplanis is questionable. First of all, the new genus is based on molecular data only and lacks any phenotypic features that differentiate it from the sister genus Microglena. Moreover, phylogeny revealing Chloroplanis lineage was based not on a full-length 18S rDNA sequence comparisons usually used to analyze genus level relationships among green algae but on a concatenated alignment of V4 domain of 18S (554 bp) and ITS rDNA, highly variable species level markers. In this tree Chloroplanis and Microglena formed a well-supported clade where Chloroplanis subclade attained moderate support but Microglena was resolved only topologically. It should be noted that the support for Chloroplanis lineage is not indisputable and it was resolved with significance in in phylogeny by Nakada et al. (2018) but not in the tree presented by Demchenko et al. (2012). Thus, Chloroplanis and Microglena could be separate genera or, equally, parts of the same genus unless there is a strong evidence that these two lineages differ phenotypically. The only feature that differentiates Chloroplanis and Microglena according to this manuscript is their distribution, marine and freshwater, respectively.
It well could be that Chloroplanis indeed represents distinct genus but this conclusion should be based on a solid molecular data and thorough analyses of morphological data and life cycle features.
Definitely the manuscript needs language improvement.
There are some minor points in the manuscript that raised questions.
Lines 101-103: our molecular markers (ITS-2 rDNA, a fragment of tufA gene, a portion of rbcL gene and the V4 hypervariable region of 18S rDNA) were amplified for further phylogenetic analyses.
Neither tufa, nor rbcL were used in the analyses. What for procedure of these sequences’ amplification was described (Lines 101-115)?
Lines 157-159: Alternative topologies were evaluated in IQTree using the RELL method [33], Kishino-Hasegawa test [34], Shimodaira-Hasegawa test [35], expected likelihood weights [36] and approximately unbiased (AU) test [37] with 10000 resamplings."
There is no data on these tests results in the text.
ITS2 sequence was 1241 bp long (Line 122) but the alignment (ITS-2.aln) for this marker was only 779 bp (Line 136). Why the alignment included a bit more than a half of the spacer? ITS-2.aln was 779 bp with or without gaps? According to the Results “We amplified …. 561 bases for the ITS-2 region” (Line 215). How long ITS2 was?
Lines 180-183: The regions selected for DNA barcode consisted of the first 15 bp in 5.8S-LSU stem, the first 5 bp of Helix 1, the first 11 bp of Helix II and the alignable base-pairs of Helix III. The extracted portions were aligned again in MARNA and then manually refined.
DNA barcode regions generally are the most conserved domains of ITS2, yet their homology assessment (=alignment) between species requires ITS2 secondary structure analysis. Are the authors sure that manual refinement (Line 183) did not introduce false phylogenetic signal? What were the criteria for this refinement?
Line 231: Bootstrap and aLRT support ≥ 50% and posterior probabilities ≥ 0.70.
According to the IQ-Tree website (http://www.iqtree.org/doc/Frequently-Asked-Questions) “One would typically start to rely on the clade if its SH-aLRT >= 80%...”. Also, usually posterior probabilities below 0.95 are disregarded and show as 1.00 not 1 (Figure 2).
Lines 218, 220 and further on: sh-alrt should be SH-aLRT
Line 245: ….(Figure 4), which supports the view, with a reasonable probability (about 76%), that these microalgae belong to the same species [13].
How this probability was estimated?
Table 1 should be reformatted. It is not readable when printed.
Authors should pay more attention to the figures. Figures 6a,b are of a very poor quality. They do not illustrate phenotypic features of the alga described in the text. Confocal microscopy was used in this study but why authors did not illustrate chloroplast morphology with the confocal images? Usually they are excellent in showing chloroplast shape, structure, etc. Authors should use LM chloroplast images in Demchenko et al., 2012 as an example.
Line 280: ….the cells were characterized by the presence of bulges, emerging from the cell surface (Figures 6b-d).
Indeed, figure 6c shows such bulges, but strangely enough they are not evident in the Fig. 6d and TEM images (Fig. 7a,b,c,f). In my opinion explanation of the bulge structure based on fig. 7c is not convincing. It is not clear what kind of “electrondense inclusions protruding from beneath the cell surface (Line 292)” to form the bulge. This morphological feature of the new alga is rather unusual and authors should make clear that it is not a result of sample preparation, etc. Perhaps DIC microscopy could confirm its presence in the cell or cell wall? If the bulges typical for Chloroplanis antarctica indeed, this feature should be compared with the cell-wall decorations in other volvocalean algae.
Line 185: old ‘Monadina clade’. Former ‘Monadina clade’?
Line 484: 18s should be 18S
Author Response
Dear reviewer 3,
We are grateful to your observations and comments. We have made new phylogenetic analyses based on the full-length 18S rDNA sequences, generated for our strain IMA076A and retrived in the Genbank achives for the other species analysized, and on the ITS-2 sequences. We provided the concatenated ML tree in the text of the manuscript and the single marker trees (18S rDNA and ITS-2) in the supplementary material. Furthermore, we evaluated alternative topologies in IQTree for our concatenated tree. The results are reported in the manuscript (Table 1).
In short, in light of the results of our new analysies, we considered the genus Microglena as divided in three subclades (two marine caldes and one freshwater lineage). ‘Clade I’ included M. antarctica and M. redcarensis, ‘clade II’ included M. reginae and M. uva-maris and ‘clade III’ included freshwater species. The monophyletic origin of the marine Microglena species is supported topologically, but further analyses are required to establish if this group correspond to a new genus or not (for example, increasing taxon sequencing may affect these results).
In conclusion, we believe that our molecular results support the identification of a new Antarctic species within the genus Microglena. We describe this specie as Microglena antarctica sp. nov., whereas alternate hypotheses on the evolution of marine and Antarctic species should be explored in the future.
Here are our point-to-point responses to your comments:
“The manuscript under review presents new genus of green algae revealed in green-ice sample collected at Inexpressible Island (Antarctica). Authors isolated this alga in the culture and studied it using molecular phylogenetic methods, light, confocal, scanning electron and transmission electron microscopy methods. Based on the molecular data, phenotypic features observed in the Antarctic alga and ecology (marine habitat) authors conclude that it represents a new genus described as Chloroplanis gen. nov. Beside Chloroplanis antarctica (newly described type species) they transfer to the new genus three species, formerly members of the genus Microglena (Chloroplanis uva-maris, Chloroplanis reginae, and Chloroplanis redcarensis) and showed that a number of strains in the culture collections also represent C. antarctica”.
1) In my opinion C. antarctica is a new species indeed and deserves description. However, entity of the new genus Chloroplanis is questionable. First of all, the new genus is based on molecular data only and lacks any phenotypic features that differentiate it from the sister genus Microglena. Moreover, phylogeny revealing Chloroplanis lineage was based not on a full-length 18S rDNA sequence comparisons usually used to analyze genus level relationships among green algae but on a concatenated alignment of V4 domain of 18S (554 bp) and ITS rDNA, highly variable species level markers. In this tree Chloroplanis and Microglena formed a well-supported clade where Chloroplanis subclade attained moderate support but Microglena was resolved only topologically. It should be noted that the support for Chloroplanis lineage is not indisputable and it was resolved with significance in in phylogeny by Nakada et al. (2018) but not in the tree presented by Demchenko et al. (2012). Thus, Chloroplanis and Microglena could be separate genera or, equally, parts of the same genus unless there is a strong evidence that these two lineages differ phenotypically. The only feature that differentiates Chloroplanis and Microglena according to this manuscript is their distribution, marine and freshwater, respectively. It well could be that Chloroplanis indeed represents distinct genus but this conclusion should be based on a solid molecular data and thorough analyses of morphological data and life cycle features.
RESPONSE 1) We modified the paper as suggested. In this sense, we performed new phylogenetic analyses including the 18S rDNA full-sequences. According to these results we can state that the genus Microglena is divided in three subclades. One of these clades includes our strain IMA076A which we describe as Microglena antarctica with a robust molecular support (ML phylogeny, species delimitation methods, comparative analyses of the secondary structures of ITS-2 and compensatory base changes). The lack of morphological differences is not a reliable indicator for excluding the possibility that marine Microglena species belong to different genus, as reported for many cryptic species and genera of microalgae. However, we understand your doubts and, for this reason, we only provide some some hypotheses on the adaptive radiation of the Microglena-clade in the discussion section.
Definitely the manuscript needs language improvement. There are some minor points in the manuscript that raised questions.
2) Lines 101-103: our molecular markers (ITS-2 rDNA, a fragment of tufA gene, a portion of rbcL gene and the V4 hypervariable region of 18S rDNA) were amplified for further phylogenetic analyses. Neither tufa, nor rbcL were used in the analyses. What for procedure of these sequences’ amplification was described (Lines 101-115)?
RESPONSE 2) We deleted this part in the text as suggested by other reviewers.
3) Lines 157-159: Alternative topologies were evaluated in IQTree using the RELL method [33], Kishino-Hasegawa test [34], Shimodaira-Hasegawa test [35], expected likelihood weights [36] and approximately unbiased (AU) test [37] with 10000 resamplings." There is no data on these tests results in the text.
RESPONSE 3) We reported these data in the text (Table 1). Thanks for the suggestion.
4) ITS2 sequence was 1241 bp long (Line 122) but the alignment (ITS-2.aln) for this marker was only 779 bp (Line 136). Why the alignment included a bit more than a half of the spacer? ITS-2.aln was 779 bp with or without gaps? According to the Results “We amplified …. 561 bases for the ITS-2 region” (Line 215). How long ITS2 was?
RESPONSE 4) We modified the text in order to better explain this part. ITS-2 and its flanking regions seqeunced here were 1241 bp long. The ITS-2 alignement (including gaps) was 779 bp long, while the lenght of IMA076A ITS-2 was 561 bp.
5) Lines 180-183: The regions selected for DNA barcode consisted of the first 15 bp in 5.8S-LSU stem, the first 5 bp of Helix 1, the first 11 bp of Helix II and the alignable base-pairs of Helix III. The extracted portions were aligned again in MARNA and then manually refined.DNA barcode regions generally are the most conserved domains of ITS2, yet their homology assessment (=alignment) between species requires ITS2 secondary structure analysis. Are the authors sure that manual refinement (Line 183) did not introduce false phylogenetic signal? What were the criteria for this refinement?
RESPONSE 5) With ‘manual refinement’ we meant the deletion of ITS-2 flanking regions (not included in the barcode). We modified the sentence to according to your suggestion.
6) Line 231: Bootstrap and aLRT support ≥ 50% and posterior probabilities ≥ 0.70. According to the IQ-Tree website (http://www.iqtree.org/doc/Frequently-Asked-Questions) “One would typically start to rely on the clade if its SH-aLRT >= 80%...”. Also, usually posterior probabilities below 0.95 are disregarded and show as 1.00 not 1 (Figure 2).
RESPONSE 6) That is true for the SH-aLRT, thanks for the correction. However PP were calculated in MrBayes, which are more reliable than the PP calculated in IQTREE. As largely reported in the literature, PP (calculated in MrBayes) ≥ 0.70 are shown in the trees.
7) Lines 218, 220 and further on: sh-alrt should be SH-aLRT
RESPONSE 7) Recommendation accepted.
8) Line 245: ….(Figure 4), which supports the view, with a reasonable probability (about 76%), that these microalgae belong to the same species [13]. How this probability was estimated?
RESPONSE 8) This is reported in the study we cited: “Coleman, A.W. The Significance of a Coincidence between Evolutionary Landmarks Found in Mating Affinity and a DNA Sequence. Protist 2000, 151, 1–9, doi:10.1078/1434-4610-00002”.
9) Table 1 should be reformatted. It is not readable when printed.
RESPONSE 9) Recommendation accepted. We moved former Table 1 to the supplementary material section (now Table S2).
10) Authors should pay more attention to the figures. Figures 6a,b are of a very poor quality. They do not illustrate phenotypic features of the alga described in the text. Confocal microscopy was used in this study but why authors did not illustrate chloroplast morphology with the confocal images? Usually they are excellent in showing chloroplast shape, structure, etc. Authors should use LM chloroplast images in Demchenko et al., 2012 as an example.
RESPONSE 10) We have modified the imagines regarding the morphological observations. Now the imagines are reported in only one Figure (Figure 5). In particular, we have replaced Figure 6a with an image obtained at the differential interference contrast microscope (DIC), where it is clear the presence of pyrenoid located at the basal part of the cup-shaped chloroplast. According to Demchenko et al.2012, the chloroplast with pyrenoid located in basal position of the chloroplast lead to ascribe our strain to the morphotype I.
11) Line 280: ….the cells were characterized by the presence of bulges, emerging from the cell surface (Figures 6b-d). Indeed, figure 6c shows such bulges, but strangely enough they are not evident in the Fig. 6d and TEM images (Fig. 7a,b,c,f). In my opinion explanation of the bulge structure based on fig. 7c is not convincing. It is not clear what kind of “electrondense inclusions protruding from beneath the cell surface (Line 292)” to form the bulge. This morphological feature of the new alga is rather unusual and authors should make clear that it is not a result of sample preparation, etc. Perhaps DIC microscopy could confirm its presence in the cell or cell wall? If the bulges typical for Chloroplanis antarctica indeed, this feature should be compared with the cell-wall decorations in other volvocalean algae.
RESPONSE 11) The bulges emerging from the cell surface could probably be due to the presence of substances accumulated in the cells. I can confirm the presence of these structures in this microalga also in environmental samples collected by me in Inexpressible Island last summer (January 2022) during the XXXVII Italian Expedition in Antarctica. Thus, I think that these structures don’t represent an artefact due to not appropriate culture conditions, because they were present also in samples collected from the lagoon and observed without fixation.
12) Line 185: old ‘Monadina clade’. Former ‘Monadina clade’?
RESPONSE 12) Recommendation accepted.
13) Line 484: 18s should be 18S
RESPONSE 13) Recommendation accepted.
Reviewer 4 Report
In this study, the authors propose to establish Chloroplanis gen. nov. and Chloroplanis antarctica sp. nov., based on a polyphasic approach. The new genus contains several marine species formerly classified as the genus Microglena, and the new species contains the strain ICE-L, which has been studied genetically, biochemically, and physiologically in recent studies. It is important to describe species and genera in order to advance our understanding of biodiversity and ecosystems. However, I found several issues in the manuscript which should be cleared before the publication.
First issue is morphological observation. Although the authors mainly rely on the molecular data for the taxonomic identification, morphological data of at least some quality should be presented in the manuscript. I recommend including both LM and TEM of a longitudinal section of the cell, showing a centrally located nucleus and a cup-shaped chloroplast with a basally positioned pyrenoid in the revised manuscript. Figure 6a should be replaced because it has too much contrast and does not explain any intracellular structure at all; in fact, it even shows a morphotype II pyrenoid-like structures (especially that in Microglena basinucleata; Demchenko et al. 2012, Eur. J. Phycol.) in the anterior third of the cell.
In addition, the authors described the presence of bulges emerging from the cell surface by electron-dense inclusions (Figs 6c and 7a–c). However, such structures are not seen in the strain ICE-L (Liu et al. 2006, Phycologia; Zhang et al. 2020, Curr. Biol.). In my experience, such structures appear in chlamydomonad cells when culture conditions are not appropriate. Based on the previous studies of the strain ICE-L and the related Microglena redcarensis, I think that the light intensity may be insufficient (~6 μmol photons/m2·s vs. 40 (ICE-L) or 30–100 (M. redcarensis) μmol photons/m2·s), or light:dark cycle may be required. Also, the TEM of the eyespot (Fig. 7f) is not a good section, and it is questionable whether it shows the exact number of the layers of eyespot globules. It should be replaced with a photo like that of the strain ARC (fig. 1H in Edie et al. 2008, Mar. Ecol. Prog. Ser.).
Second issue is phylogenetic analyses. The present phylogenetic result is one of the rationales for the establishment of the new genus. However, is it appropriate to combine 18S rDNA and ITS-2, which have completely different evolutionary rates? I think that most of the informative sites in the molecular phylogeny is in the ITS-2 region, since the authors use only V4 region of the 18S rDNA. I would like to see an rbcL-based phylogenetic tree; however, as the authors point out, the rbcL sequences of Microglena or Chloroplanis are largely unavailable (neglect of the previous studies!). Therefore, I recommend showing phylogenetic results based only on 18S rDNA or ITS-2 to the readers as supplemental data. Also, please conduct a phylogenetic analysis using longer (more than 1500 bp) 18S rDNA sequences, just for the available strains/species.
Although freshwater and marine species are physiologically very different, I am concerned that the monophyly of the Microglena species is not statistically supported in the Figure 2. Several previous phylogenetic results based on 18S rDNA (e.g., Demchenko et al. 2012; Nakada et al. 2018, Phycol. Res.) suggested the paraphyly of the species/strains which are classified as Chloroplanis in this study. In addition, the statistical support for the monophyly of Chloroplanis is not high (it is well known that PP=0.99 is not robust). Possibly, long branch attraction may occur. Therefore, I would also like to see the phylogenetic result based on the method against long branch attraction, such as CAT+GTR in PhyloBayes. By the way, I think that the chaetophoralean OTUs used as the outgroup are too phylogenetically distant from ‘Monadina’-clade and not suitable as the outgroup.
Finally, the manuscript contains several misunderstandings regarding the nomenclatural rules for microalgae. This is crucial for proposing a new species and new combinations; please revise them based on my comments (see below).
Followings are the comments listed chronologically.
Line 24. “IMA076A as the new species …” should be revised as “IMA06A and its relatives as the new species …”.
Line 45–52. Since the journal is not an algae journal, I think more explanations of the genus Chlamydomonas (especially its morphological characteristics) are required in this section for readers.
Lines 46 and 51. Insert the citation numbers for Ehrenberg (1834), Demchenko et al. (2012), and Ehrenberg (1832), respectively.
Line 67. “the multi-gene phylogenetic analyses” should be replaced with “the phylogenetic analysis using concatenated 18S rDNA and ITS-2 sequences”, since ITS-2 is not a gene.
Lines 82 and 86. Generally, “F/2” are noted as “f/2”.
Line 94 (Figure 1). Align the orientation of the direction of the inserted figures.
Line 104. Insert comma between “[16]” and “ITS4”.
Line 153. Is “sampling trees every 1000 generations” true? I think “every 100 generations” is correct.
Line 232 (Fig. 2). Microglena “skujae” SAG 16.90 was re-classified as M. media (Nakada et al. 2014, Acta Phytotax. Geobot.). If the authors treated that the strain SAG 16.90 is separated from M. media strain HcCl-5-3, the authors must propose a new combination, Microglena nova, since M. “skujae” is a superfluous name (see Nakada et al. 2014).
Line 316. Generally, initials of taxon authors should be geven as necessary according to the author information in the International Plant Name Index (https://www.ipni.org/) [see Recommendation 46A and its Note of the International Code of Nomenclature for algae, fungi, and plants (ICN) (Shenzhen Code)].
Line 336–339 (description for C. antarctica). For readability, the diagnosis of the new species should be placed just after the new species name. Thus, this description should be transferred just after DNA Sequence Available (Lines 358–359). In addition, it would be better to rename “Description” to “Remarks” based on the content of the paragraph.
Line 349–350. Delete “(designated here)”. This is for the designation of lectotypes, neotypes, and epitypes (see Article 7.11 and its Note of the ICN), although several researchers confuse it. In addition, indicate the name of the sample (strain number) used to create the holotype. Moreover, I strongly encourage the authors to deposit the strain used for the holotype of C. antarctica in some public culture collection (e.g CCCryo, SAG, etc.). This is very important for future research.
Line 352. Other habitats of this new species (e.g., ICE-L and CCCryo 309-05) can be listed.
Line 358. “18s” should be revised as “18S”.
Line 361–410 (proposing new combinations). These paragraphs must be revised thoroughly. For proposing a new combination without any additional work by the authors (e.g. emendation of diagnosis), it is sufficient to clearly indicate its basionym (or replaced synonym) and a full and direct reference given to its author and place of valid publication, with page or plate reference and date [see Article 41 (especially 41.5 and its note) of the ICN]. Synonyms can be listed with the basionym; but they must be distinguished (see Demchenko et al. 2012; Nakada et al. 2014). Since the authors did NOT designate the holotype(!), lectotype, epitype for Chlamydomonas uva-maris, Chlamydomonas reginae, and Microglena redcarensis (basionyms of the new combinations in the manuscript), lines 371–378, 388–395, and 404–410 of the manuscript are completely superfluous and must be deleted. Followings are the example of the revised proposal of the new combinations.
******************************
Chloroplanis uva-maris (Butcher) Trentin, Negrisolo, Moschin, Veronese, Cecchetto & I. Moro comb. nov.
Basionym: Chlamydomonas uva-maris Butcher (1959). Smaller Algae of British Coastal Waters. Part I: Introduction and Chlorophyceae. Fisheries Invest, ser. 4: 53.
Synonym: Microglena uva-maris (Butcher) Demchenko, Mikhailyuk, Coleman & Pröschold (2012). Eur. J. Phycol., 47: 287.
Chloroplanis reginae (H. Ettl & J.C. Green) Trentin, Negrisolo, Moschin, Veronese, Cecchetto & I. Moro comb. nov.
Basionym: Chlamydomonas reginae H. Ettl & J.C. Green (1973). J. Mar. Biol. Assoc. UK, 53: 975.
Synonym: Microglena reginae (H. Ettl & J.C. Green) Demchenko, Mikhailyuk, Coleman & Pröschold (2012). Eur. J. Phycol., 47: 287.
Chloroplanis redcarensis (Nakada & S. Takahashi) Trentin, Negrisolo, Moschin, Veronese, Cecchetto & I. Moro comb. nov.
Basionym: Microglena redcarensis Nakada & S. Takahashi (2018). Phycol. Res., 66: 311.
******************************
Line 454–467. I don’t know the style of this journal, but Conclusion section is generally set when the Results and Discussion sections are merged. Thus, I think this section should be merged with the Discussion section.
Appendix A (Table A1). Revise it based on the following comments and confirm it thoroughly again.
18s > 18S.
ITS-2 of Aphanochaete magnum: AF182816 > HQ646383.
Microglena media:
HcCl-5-3 > HcCl-5-3 (= NIES-2743).
ITS-2: AB694003 > AB872922.
Microglena opysthopyren
TkS0811B3 > TkS0811B3 (= NIES-2744)
ITS-2: AB694004 > AB872921.
Microglena redcarensis > Chloroplanis redcarensis.
ITS-2: LC214871 > LC214872.
Strain # of Phacotus lenticularis: 97/2 SAG 16.99 > SAG 16.99.
Line 485–605 (References). Confirm the references thoroughly again. Since the Journal of Phycology use unique style, the last of the title frequently has superfluous “1” (e.g. Lines 496 and 500).
Author Response
Dear reviewer 4,
We are grateful to your observations and comments. We have made new phylogenetic analyses based on the full-length 18S rDNA sequences, generated for our strain IMA076A and retrived in the Genbank achives for the other species analysized, and on the ITS-2 sequences. We provided the concatenated ML tree in the text of the manuscript and the single marker trees (18S rDNA and ITS-2) in the supplementary material. Furthermore, we evaluated alternative topologies in IQTree for our concatenated tree. The results are reported in the manuscript (Table 1).
In short, in light of the results of our new analysies, we considered the genus Microglena as divided in three subclades (two marine caldes and one freshwater lineage). ‘Clade I’ included M. antarctica and M. redcarensis, ‘clade II’ included M. reginae and M. uva-maris and ‘clade III’ included freshwater species. The monophyletic origin of the marine Microglena species is supported topologically, but further analyses are required to establish if this group correspond to a new genus or not (for example, increasing taxon sequencing may affect these results).
In conclusion, we believe that our molecular results support the identification of a new Antarctic species within the genus Microglena. We describe this specie as Microglena antarctica sp. nov., whereas alternate hypotheses on the evolution of marine and Antarctic species should be explored in the future.
Here are our point-to-point responses to your comments:
“In this study, the authors propose to establish Chloroplanis gen. nov. and Chloroplanis antarctica sp. nov., based on a polyphasic approach. The new genus contains several marine species formerly classified as the genus Microglena, and the new species contains the strain ICE-L, which has been studied genetically, biochemically, and physiologically in recent studies. It is important to describe species and genera in order to advance our understanding of biodiversity and ecosystems. However, I found several issues in the manuscript which should be cleared before the publication”.
1) First issue is morphological observation. Although the authors mainly rely on the molecular data for the taxonomic identification, morphological data of at least some quality should be presented in the manuscript. I recommend including both LM and TEM of a longitudinal section of the cell, showing a centrally located nucleus and a cup-shaped chloroplast with a basally positioned pyrenoid in the revised manuscript. Figure 6a should be replaced because it has too much contrast and does not explain any intracellular structure at all; in fact, it even shows a morphotype II pyrenoid-like structures (especially that in Microglena basinucleata; Demchenko et al. 2012, Eur. J. Phycol.) in the anterior third of the cell. n addition, the authors described the presence of bulges emerging from the cell surface by electron-dense inclusions (Figs 6c and 7a–c). However, such structures are not seen in the strain ICE-L (Liu et al. 2006, Phycologia; Zhang et al. 2020, Curr. Biol.). In my experience, such structures appear in chlamydomonad cells when culture conditions are not appropriate. Based on the previous studies of the strain ICE-L and the related Microglena redcarensis, I think that the light intensity may be insufficient (~6 μmol photons/m2·s vs. 40 (ICE-L) or 30–100 (M. redcarensis) μmol photons/m2·s), or light:dark cycle may be required. Also, the TEM of the eyespot (Fig. 7f) is not a good section, and it is questionable whether it shows the exact number of the layers of eyespot globules. It should be replaced with a photo like that of the strain ARC (fig. 1H in Edie et al. 2008, Mar. Ecol. Prog. Ser.).
RESPONSE 1) We have modified the imagines regarding the morphological observations. Now the imagines are reported in only one Figure (Figure 5). In particular, we have replaced Figure 6a with an image obtained at the differential interference contrast microscope (DIC), as suggested by the reviewer 3, where it is clear the presence of pyrenoid located at the basal part of the cup-shaped chloroplast. According to Demchenko et al.2012, the chloroplast with pyrenoid located in basal position of the chloroplast lead to ascribe our strain to the morphotype I.
The bulges emerging from the cell surface could probably be due to the presence of substances accumulated in the cells. I can confirm the presence of these structures in this microalga also in environmental samples collected by me in Inexpressible Island last summer (January 2022) during the XXXVII Italian Expedition in Antarctica. Thus, I think that these structures don’t represent an artefact due to not appropriate culture conditions, because they were present also in samples collected from the lagoon and observed without fixation.
Regarding the eyespot, we agree with and thus we have decided to remove the TEM micrograph, because it was not a good section.
2) Second issue is phylogenetic analyses. The present phylogenetic result is one of the rationales for the establishment of the new genus. However, is it appropriate to combine 18S rDNA and ITS-2, which have completely different evolutionary rates? I think that most of the informative sites in the molecular phylogeny is in the ITS-2 region, since the authors use only V4 region of the 18S rDNA. I would like to see an rbcL-based phylogenetic tree; however, as the authors point out, the rbcL sequences of Microglena or Chloroplanis are largely unavailable (neglect of the previous studies!). Therefore, I recommend showing phylogenetic results based only on 18S rDNA or ITS-2 to the readers as supplemental data. Also, please conduct a phylogenetic analysis using longer (more than 1500 bp) 18S rDNA sequences, just for the available strains/species.
RESPONSE 2) It is appropriate to combine 18S rDNA and ITS-2 if considering different evolutionary models. For this reasons our concatenated dataset was analyzed using partitions, as reported by most of multi-gene analyses in literature, such as the method adopted by Demchenko et al., 2012 where the genus Microglena was established. We provided the concatenated ML tree in the text of the manuscript and the single marker trees (18S rDNA and ITS-2) in the supplementary material.
3) Although freshwater and marine species are physiologically very different, I am concerned that the monophyly of the Microglena species is not statistically supported in the Figure 2. Several previous phylogenetic results based on 18S rDNA (e.g., Demchenko et al. 2012; Nakada et al. 2018, Phycol. Res.) suggested the paraphyly of the species/strains which are classified as Chloroplanis in this study. In addition, the statistical support for the monophyly of Chloroplanis is not high (it is well known that PP=0.99 is not robust). Possibly, long branch attraction may occur. Therefore, I would also like to see the phylogenetic result based on the method against long branch attraction, such as CAT+GTR in PhyloBayes. By the way, I think that the chaetophoralean OTUs used as the outgroup are too phylogenetically distant from ‘Monadina’-clade and not suitable as the outgroup.
RESPONSE 3) The monophyly of Microglena species is statistically supported whene combining 18S rDNA and ITS-2 sequences, as reported in Demchenko et al. 2012 and in this study (98/1/98, approximate Likelihood Ratio Test (aLRT) values, Bayesian Posterior Probabilities (PP) and ML bootstrap values). According to these results we can state that the genus Microglena is divided in three subclades. One of these clades includes our strain IMA076A which we describe as Microglena antarctica with a robust molecular support (ML phylogeny, species delimitation methods, comparative analyses of the secondary structures of ITS-2 and compensatory base changes). Regarding the outgroup, we realied again on the ML phylogeny reported in Demchenko et al. 2012.
4) Finally, the manuscript contains several misunderstandings regarding the nomenclatural rules for microalgae. This is crucial for proposing a new species and new combinations; please revise them based on my comments (see below).
Followings are the comments listed chronologically.
Line 24. “IMA076A as the new species …” should be revised as “IMA06A and its relatives as the new species …”.
RESPONSE 4) Recommendation accepted.
5) Line 45–52. Since the journal is not an algae journal, I think more explanations of the genus Chlamydomonas (especially its morphological characteristics) are required in this section for readers.
RESPONSE 5) We do not aim to spend much word regarding the morphology of this lineage since it consists of cryptics species which are not possible to distinguish basing on solely morphological features.
6) Lines 46 and 51. Insert the citation numbers for Ehrenberg (1834), Demchenko et al. (2012), and Ehrenberg (1832), respectively.
RESPONSE 6) Recommendation accepted.
7) Line 67. “the multi-gene phylogenetic analyses” should be replaced with “the phylogenetic analysis using concatenated 18S rDNA and ITS-2 sequences”, since ITS-2 is not a gene.
RESPONSE 7) Recommendation accepted.
8) Lines 82 and 86. Generally, “F/2” are noted as “f/2”.
RESPONSE 8) Recommendation accepted.
9) Line 94 (Figure 1). Align the orientation of the direction of the inserted figures.
RESPONSE 9) Recommendation accepted.
10) Line 104. Insert comma between “[16]” and “ITS4”.
RESPONSE 10) Recommendation accepted.
11) Line 153. Is “sampling trees every 1000 generations” true? I think “every 100 generations” is correct.
RESPONSE 11) Recommendation accepted.
12) Line 232 (Fig. 2). Microglena “skujae” SAG 16.90 was re-classified as M. media (Nakada et al. 2014, Acta Phytotax. Geobot.). If the authors treated that the strain SAG 16.90 is separated from M. media strain HcCl-5-3, the authors must propose a new combination, Microglena nova, since M. “skujae” is a superfluous name (see Nakada et al. 2014).
RESPONSE 12) Recommendation accepted.
13) Line 316. Generally, initials of taxon authors should be geven as necessary according to the author information in the International Plant Name Index (https://www.ipni.org/) [see Recommendation 46A and its Note of the International Code of Nomenclature for algae, fungi, and plants (ICN) (Shenzhen Code)].
RESPONSE 13) Recommendation accepted.
14) Line 336–339 (description for C. antarctica). For readability, the diagnosis of the new species should be placed just after the new species name. Thus, this description should be transferred just after DNA Sequence Available (Lines 358–359). In addition, it would be better to rename “Description” to “Remarks” based on the content of the paragraph.
RESPONSE 14) Recommendation accepted.
15) Line 349–350. Delete “(designated here)”. This is for the designation of lectotypes, neotypes, and epitypes (see Article 7.11 and its Note of the ICN), although several researchers confuse it. In addition, indicate the name of the sample (strain number) used to create the holotype. Moreover, I strongly encourage the authors to deposit the strain used for the holotype of C. antarctica in some public culture collection (e.g CCCryo, SAG, etc.). This is very important for future research.
RESPONSE 15) Recommendation accepted. Regarding the deposition of the strain in some public culture collection (e.g CCCryo, SAG, etc.), we agree with you and usually we deposit the strains in public culture collections, but the policy on the deposition of Antarctic samples in public collection is in the process of deciding by Programma Nazionale Ricerche in Antartide (PNRA). For now, we maintain the holotype in the Italian National Antarctic Museum (MNAIT, Section of Genoa), but at a later date we will decide to deposit it in a public culture collection.
16) Line 352. Other habitats of this new species (e.g., ICE-L and CCCryo 309-05) can be listed.
RESPONSE 16) We only provide the information regarding our strain to avoid misunderstanding.
17) Line 358. “18s” should be revised as “18S”.
RESPONSE 17) Recommendation accepted.
18) Line 361–410 (proposing new combinations). These paragraphs must be revised thoroughly. For proposing a new combination without any additional work by the authors (e.g. emendation of diagnosis), it is sufficient to clearly indicate its basionym (or replaced synonym) and a full and direct reference given to its author and place of valid publication, with page or plate reference and date [see Article 41 (especially 41.5 and its note) of the ICN]. Synonyms can be listed with the basionym; but they must be distinguished (see Demchenko et al. 2012; Nakada et al. 2014). Since the authors did NOT designate the holotype(!), lectotype, epitype for Chlamydomonas uva-maris, Chlamydomonas reginae, and Microglena redcarensis (basionyms of the new combinations in the manuscript), lines 371–378, 388–395, and 404–410 of the manuscript are completely superfluous and must be deleted. Followings are the example of the revised proposal of the new combinations.
******************************
Chloroplanis uva-maris (Butcher) Trentin, Negrisolo, Moschin, Veronese, Cecchetto & I. Moro comb. nov.
Basionym: Chlamydomonas uva-maris Butcher (1959). Smaller Algae of British Coastal Waters. Part I: Introduction and Chlorophyceae. Fisheries Invest, ser. 4: 53.
Synonym: Microglena uva-maris (Butcher) Demchenko, Mikhailyuk, Coleman & Pröschold (2012). Eur. J. Phycol., 47: 287.
Chloroplanis reginae (H. Ettl & J.C. Green) Trentin, Negrisolo, Moschin, Veronese, Cecchetto & I. Moro comb. nov.
Basionym: Chlamydomonas reginae H. Ettl & J.C. Green (1973). J. Mar. Biol. Assoc. UK, 53: 975.
Synonym: Microglena reginae (H. Ettl & J.C. Green) Demchenko, Mikhailyuk, Coleman & Pröschold (2012). Eur. J. Phycol., 47: 287.
Chloroplanis redcarensis (Nakada & S. Takahashi) Trentin, Negrisolo, Moschin, Veronese, Cecchetto & I. Moro comb. nov.
Basionym: Microglena redcarensis Nakada & S. Takahashi (2018). Phycol. Res., 66: 311.
******************************
RESPONSE 18) We modified the text describing IMA076A and its relatives as Microglena antarctica, so we do not need to propose new combinations.
19) Line 454–467. I don’t know the style of this journal, but Conclusion section is generally set when the Results and Discussion sections are merged. Thus, I think this section should be merged with the Discussion section.
RESPONSE 19) There is not a general rule.
20) Appendix A (Table A1). Revise it based on the following comments and confirm it thoroughly again.
18s > 18S.
ITS-2 of Aphanochaete magnum: AF182816 > HQ646383.
Microglena media:
HcCl-5-3 > HcCl-5-3 (= NIES-2743).
ITS-2: AB694003 > AB872922.
Microglena opysthopyren
TkS0811B3 > TkS0811B3 (= NIES-2744)
ITS-2: AB694004 > AB872921.
Microglena redcarensis > Chloroplanis redcarensis.
ITS-2: LC214871 > LC214872.
Strain # of Phacotus lenticularis: 97/2 SAG 16.99 > SAG 16.99.
RESPONSE 20) Recommendation accepted.
21) Line 485–605 (References). Confirm the references thoroughly again. Since the Journal of Phycology use unique style, the last of the title frequently has superfluous “1” (e.g. Lines 496 and 500).
RESPONSE 21) Recommendation accepted.
Round 2
Reviewer 3 Report
In the revised manuscript authors have taken into consideration most questions, comments, and suggestions made by the reviewer. Namely, they obtained and used in the phylogenetic analyses full-length 18S rDNA sequence, reconsidered results of these analyses and morphological observations and refrained from describing new genus. Instead they described a new species Microglena antarctica. Indeed, this species occupies distinct position in the Microglena clade I and represented by multiple accessions. Unfortunately, authors could not provide any morphological data that support this placement perhaps except presence of bulges/granulation on the cell-wall.
Authors improved figures quality by adding some new LM and removing non-informative TEM images. I found interesting that in alga shown in the Fig. 6a chloroplast is clearly detached from the posterior end of the cell and the same feature is evident in the Fig. 6d but absent in illustrations of other Microglena species presented by Demchenko et al., 2012. I wonder whether this morphological feature is typical for M. antarctica or not? If it is characteristic for the species, perhaps that could be an additional distinctive character.
Unfortunately, in the revised manuscript I have noticed a few points that require clarification. 1. The 18S sequence obtained for the strain IMA076A was 1781 bp long (Line 137) and the alignment based on that marker had the same length (Line 152). It is rather unlikely that other sequences contained no insertions or places with ambiguous homology and no gaps were inserted during the alignment process. 2. Figure 2 combines results of the phylogenetic analyses and species delimitation analyses. I found it strange that all species delimitation methods did not distinguish M. braunii from M. coccifera and M. opisthopyren despite strong supports for the respective clades. In contrast to that, two out of five methods recognised identical sequences of M. charkoviensis as different entities! In my opinion these strange results undermine value of the species delimitation analyses performed, especially in light of such a distinct phylogenetic position of M. antarctica. I doubt that anyone would question distinctness of this species.
The same I can say about alternative topology tests. Phylogeny presented in the Fig. 2 is well resolved and convincing, and the clades I, II, and III are well supported. In my opinion, these tests look like excessive.
Also, I suggest authors change description of the phylogenetic analyses results by describing overall topology first (split of Microglena into 3 clades) and then coming to the position of M. antarctica in the tree.
Author Response
Dear reviewer 3,
We are grateful to your observations and comments. As you suggested in the previous revision, we adopted a more cautious approach in the description of the Antarctic strain IMA076A and decided to circumscribe this strain and its relatives as a new species within the genus Microglena. Unfortunately, species of the Microglena-clade lack of diagnostic morphological characters. In this sense, the genus was described by Demchenko et al. 2012 using molecular data and especially the CBC approach proposed by Coleman (2000, 2009), Moniz & Kaczmarska (2009) and Bock et al. (2011). The presence of bulges and the adaptation to Antarctic environment are the main morpho-ecological characters that could help in the delimitation of Microglena antarctica sp. nov.
We believe that our molecular results support the identification of a new Antarctic species within the genus Microglena. We describe this specie as Microglena antarctica sp. nov., whereas alternate hypotheses on the evolution of marine and Antarctic species should be explored in the future.
Here are our point-to-point responses to your comments:
In the revised manuscript authors have taken into consideration most questions, comments, and suggestions made by the reviewer. Namely, they obtained and used in the phylogenetic analyses full-length 18S rDNA sequence, reconsidered results of these analyses and morphological observations and refrained from describing new genus. Instead they described a new species Microglena antarctica. Indeed, this species occupies distinct position in the Microglena clade I and represented by multiple accessions. Unfortunately, authors could not provide any morphological data that support this placement perhaps except presence of bulges/granulation on the cell-wall.
1) Authors improved figures quality by adding some new LM and removing non-informative TEM images. I found interesting that in alga shown in the Fig. 6a chloroplast is clearly detached from the posterior end of the cell and the same feature is evident in the Fig. 6d but absent in illustrations of other Microglena species presented by Demchenko et al., 2012. I wonder whether this morphological feature is typical for M. antarctica or not? If it is characteristic for the species, perhaps that could be an additional distinctive character.
Response 1) We agree with you that the cells seem detached from the posterior end, but we can’t affirm that this represents an additional distinctive character for this new species, because this character could represent a physiological response by this microalga to the particular environmental conditions.
2) Unfortunately, in the revised manuscript I have noticed a few points that require clarification. The 18S sequence obtained for the strain IMA076A was 1781 bp long (Line 137) and the alignment based on that marker had the same length (Line 152). It is rather unlikely that other sequences contained no insertions or places with ambiguous homology and no gaps were inserted during the alignment process.
Response 2) The alignment is 1806 bp long. It was just an oversight on our part. We corrected the text.
3) Figure 2 combines results of the phylogenetic analyses and species delimitation analyses. I found it strange that all species delimitation methods did not distinguish M. braunii from M. coccifera and M. opisthopyren despite strong supports for the respective clades. In contrast to that, two out of five methods recognised identical sequences of M. charkoviensis as different entities! In my opinion these strange results undermine value of the species delimitation analyses performed, especially in light of such a distinct phylogenetic position of M. antarctica. I doubt that anyone would question distinctness of this species.
Response 3) We corrected the species delimitation results depicted in Figure 2. In particular, not distinguishing M. braunii from M. coccifera and M. opisthopyren was just an other oversight during the manual reworking of the figure. Regarding the separation of M. charkoviensis ACKU 274-03 and ACKU 267-03 as different entities, this is not strange and it is related to the sentitivity of the method applied. Similar cases are reported in literature for other species of green microalgae, such as in the evalation of the species boundaries within the genus Coccomyxa by Darienko et al. 2015. In that study (Fig. 2), the results of different species delimitation approaches (distance-based methods: ABGD and K/θ; phylogeny-based methods: GMYC (ST = single threshold, MT = multi threshold) and PTP (ML = maximum likelihood, MB = Bayesian) were reported and it was indicated with different colours if the species delimitation provided by a specific methos was resolved or not. Overall, the results of the species delimitation analyses are not strange and support the phylogenetic position of M. antarctica. These results are corroborated by the compensatory base changes (CBCs) analysis.
Darienko, T., Gustavs, L., Eggert, A., Wolf, W., & Pröschold, T. (2015). Evaluating the Species Boundaries of Green Microalgae (Coccomyxa, Trebouxiophyceae, Chlorophyta) Using Integrative Taxonomy and DNA Barcoding with Further Implications for the Species Identification in Environmental Samples. PLOS ONE, 10(6), e0127838. doi:10.1371/journal.pone.0127838
4) The same I can say about alternative topology tests. Phylogeny presented in the Fig. 2 is well resolved and convincing, and the clades I, II, and III are well supported. In my opinion, these tests look like excessive.
Response 4) We added these tests as requested by other reviewers. We believe it could be usefull to present the result of alternative topology tests, especially in the light of the speculations on the origin of marine and freshwater species (Discussion section).
5) Also, I suggest authors change description of the phylogenetic analyses results by describing overall topology first (split of Microglena into 3 clades) and then coming to the position of M. antarctica in the tree.
Response 5) Recommendation accepted.
Reviewer 4 Report
I am very pleased that the authors revised the manuscript seriously based on the reviewers' comments. I hope that the Antarctic strain used in this study will be deposited in public culture collections and be available to many researchers. Please check the following minor comments.
Line 104–105 (Figure 1). Align the orientation of the direction of the inserted figures. I could not find any changes excluding the size of the figure.
Line 499. “SAG 18.89” should not be italicized.
Table S1. Please confirm the accession number for the ITS-2 of Aphanochaete magnum. AF182816 does not contain ITS region.
Author Response
I am very pleased that the authors revised the manuscript seriously based on the reviewers' comments. I hope that the Antarctic strain used in this study will be deposited in public culture collections and be available to many researchers. Please check the following minor comments.
1) Line 104–105 (Figure 1). Align the orientation of the direction of the inserted figures. I could not find any changes excluding the size of the figure.
Response 1) Recommendation accepted.
2) Line 499. “SAG 18.89” should not be italicized.
Response 2) Recommendation accepted.
3) Table S1. Please confirm the accession number for the ITS-2 of Aphanochaete magnum. AF182816 does not contain ITS region.
Response 3) We confirm and modified the table including the ITS-2 sequence accession number (HQ646383).